# Textual Aesthetics in Large Language Models

## Abstract

Image aesthetics is a crucial metric in the field of image generation. However, textual aesthetics has not been sufficiently explored. With the widespread application of large language models (LLMs), previous work has primarily focused on the correctness of content and the helpfulness of responses. Nonetheless, providing responses with textual aesthetics is also an important factor for LLMs, which can offer a cleaner layout and ensure greater consistency and coherence in content. In this work, we introduce a pipeline for aesthetics polishing and help construct a textual aesthetics dataset named TEXAES. We propose a textual aesthetics-powered fine-tuning method based on direct preference optimization, termed TAPO, which leverages textual aesthetics without compromising content correctness. Additionally, we develop two evaluation methods for textual aesthetics based on text and image analysis, respectively. Our experiments demonstrate that using textual aesthetics data and employing the TAPO fine-tuning method not only improves aesthetic scores but also enhances performance on general evaluation datasets such as AlpacalEval and Anera-hard.

## 1 Introduction

Image aesthetics (Huang et al., 2024a; Murray et al., 2012; Kong et al., 2016; Ke et al., 2021; Bosse et al., 2017) has emerged as a prominent research area within computer vision, focusing on assessing and improving the visual appeal of images. Aesthetics has recently been integrated into state-of-the-art image generation models, such as diffusion models (Rombach et al., 2022), significantly enhancing the visual quality of generated images (Wu et al., 2024a; 2023) and aligning them more closely with human preferences (Huang et al., 2024a; Wu et al., 2024b; 2023).

Meanwhile, advancements in large language models (LLMs) like ChatGPT (OpenAI, 2023) and LLaMA (Touvron et al., 2023b; Dubey et al., 2024) have demonstrated impressive generative capabilities across various domains, including code, articles, and web content. Although LLMs have made significant progress in generating textual content, enhancing the aesthetic quality of their output remains a critical challenge. A more aesthetically appealing and organized output not only improves user engagement by making the content more intuitive and comfortable to read but also enhances consistency and coherence. Consequently, exploring the textual aesthetics of LLMs is a highly desirable area of research.

In this work, we present the first investigation into improving the aesthetic quality of text generated by LLMs. Unlike image aesthetics benefiting from numerous large-scale aesthetic datasets (e.g., AVA (Murray et al., 2012) and AesBench (Huang et al., 2024b)), advanced aesthetic learning technology (Huang et al., 2024a; Zhang & Liu, 2023; Yang et al., 2022; Su et al., 2020) and reliable aesthetic evaluation methods (Deng et al., 2017; Su et al., 2011), textual aesthetics in LLMs lacks similar resources and established models.

To address this challenge, we first designed an aesthetic data generation pipeline leveraging GPT-4o for aesthetic polishing. This scalable pipeline can generate large volumes of high-quality aesthetic preference data. Based on this framework, we constructed the first aesthetic dataset in the LLM domain, TEXAES, which contains a total of 50,390 prompts data.

Based on TEXAES, existing post-training techniques such as DPO (Rafailov et al., 2024b) can be used to fine-tune current LLMs at the aesthetic level. However, we found that directly applying these techniques not only failed to align effectively with the characteristics of our TEXAES, limiting its impact on aesthetic fine-tuning, but also negatively impacted the overall performance of these

LLMs. To address this issue, we propose **T**extual **A**esthetics **P**reference **O**ptimization (TAPO) which employs the Plackett-Luce (Luce, 1959; Plackett, 1975) model with adjustable optimization weights to better leverage our dataset and enhance aesthetic fine-tuning performance. Furthermore, to better assess the aesthetic quality of LLM outputs, we have developed two evaluation pipelines: one based on text and the other based on images, respectively.

To validate the effectiveness of our TEXAES and TAPO, we performed aesthetic fine-tuning on the open-source LLaMA series models (Dubey et al., 2024) and compared the aesthetic scores of the fine-tuned LLMs with state-of-the-art LLMs at different scales (from 8B to 70B). Additionally, to ensure objective and reliable results, we employed human experts for professional evaluation. Extensive experimental results ultimately demonstrated the effectiveness of our TEXAES and TAPO.

Our main contributions are listed as follow:

- To the best of our knowledge, we for the first time indciate the crucial issue of exploring and improving the textual aesthetics in LLMs.
- We systematically identify the lack of related textual aesthetics datasets, and introduce a novel pipeline for aesthetic text polishing and contribute to the construction of a textual aesthetics dataset, named TEXAES.
- Based on TEXAES, we propose a DPO-based aesthetic fine-tuning algorithm, named TAPO, to effectively enhances the LLMs' aesthetic quality while preserving its general performance.
- Both qualitative and quantitative extensive experiments demonstrate that utilizing TEXAES and TAPO not only improves aesthetic scores but also enhances the general capabilities of LLMs.

## 2 RELATED WORKS

### 2.1 IMAGE AESTHETICS

Image aesthetics (Huang et al., 2024a; Murray et al., 2012; Kong et al., 2016) is a subfield of computer vision that focuses on assessing (Deng et al., 2017; Su et al., 2011) and improving the aesthetic quality of images (Bhattacharya et al., 2010; Deng et al., 2018). Early work in the field of image aesthetics focused on using handcrafted metrics to assess aesthetic scores (Nack et al., 2001; Neumann et al., 2005). However, with the development of deep learning, there has been significant interest in applying CNN (Bosse et al., 2017; Li et al., 2018; Su et al., 2020) or Transformer (Ke et al., 2021; Zhang & Liu, 2023; Yang et al., 2022; Qin et al., 2023) based methods to solve image aesthetics problems, which have demonstrated promising results. Recently, multi-modal large language models (MLLMs) have shown superior aesthetic perception and robustness in the fields of image aesthetics, greatly surpassing lightweight models due to their vast knowledge base and strong reasoning and memory capabilities. (Huang et al., 2024a;b; Wu et al., 2024b).

### 2.2 LLM PREFERENCES DATA

Preference learning is an optimization method for LLMs designed to enhance their ability to generate outputs that better align with human preferences (Fürnkranz & Hüllermeier, 2010; Schulman et al., 2017; Rafailov et al., 2024b; Ouyang et al., 2022). Increasing attention has also been drawn to the importance of data used during the preference learning phase. Some studies focus on constructing domain-specific datasets for preference learning, e.g., summarization (Stiennon et al., 2020; Wu et al., 2021) and question answering (Nakano et al., 2021). Cui et al. (2024) highlights the scarcity of large-scale, general-purpose preference datasets and propose UltraFeedback to addresses this gap by collecting over 1 million preference feedback samples using GPT-4 (OpenAI, 2023). Lee et al. (2023) also pointed out that utilizing AI-generated preference feedback is an effective and cost-efficient method for expanding preference datasets. While the aforementioned work provides preference datasets for specific domains as well as general-purpose tasks, none of them have addressed the critical area of text aesthetics in LLMs, which motivated us to design corresponding data construction pipeline and related dataset like TEXAES to support future research in text aesthetics.

## 3 TEXTUAL AESTHETICS

### 3.1 OVERVIEW

Textual aesthetics, which encompass the aesthetic attributes of a text at both the content and visual levels, can be dissected into four fundamental aspects. **Clarity** (readability) pertains to the ease with which a text can be read and comprehended, necessitating optimal sentence length and grammatical complexity (DuBay, 2004). **Layout** (visual organization) involves the systematic arrangement of text elements, such as headings and subheadings, to guide the reader effectively. **Uniformity** (consistency) demands a consistent style and formatting throughout the text to enhance readability and facilitate a smoother reading experience. **Coherence** (overall structure) ensures that paragraphs are well-organized and logically connected, facilitating easier comprehension of the content (Van Silfhout et al., 2014).

### 3.2 AESTHETICS POLISHING

Human preference data is critical for aligning large language models and improving their performance across various dimensions, such as helpfulness (Askell et al., 2021; Kreutzer et al., 2018; Stiennon et al., 2020), harmlessness (Bai et al., 2022; Glaese et al., 2022), and honesty (Ouyang et al., 2022). Consequently, we believe that a textual aesthetic preference dataset will also be beneficial for research on the alignment of LLMs. However, current literature reveals a conspicuous absence of research specifically addressing the textual aesthetics of LLMs, as well as a lack of corresponding textual aesthetic preference data. To address this gap, we have developed a method for textual aesthetic polishing to construct a dataset that optimizes the aesthetic preferences of LLMs.

Given that the goal of polishing is to enhance textual aesthetics, we can build our textual aesthetic preference dataset based on an available preference dataset such as UltraFeedback (Cui et al., 2024). UltraFeedback is a comprehensive dataset with responses evaluated by GPT-4 based on criteria such as instruction-following, honesty, and helpfulness. Since the selected data exhibits higher scores in these areas, thereby aligning more closely with human preferences, we can utilize these chosen responses as our candidates to build our textual aesthetic preference dataset.

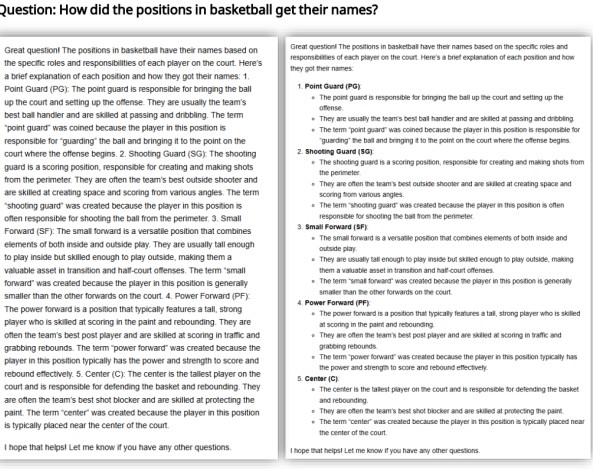

Figure 1: Comparison of responses between the UltraFeedback and TEXAES datasets.

To effectively achieve our objectives, we designed a chain of thought (Wei et al., 2023) methodology by using GPT-4o to polish our original responses, the following steps were taken:

1. **Semantic Analysis**: GPT-4o initially analyzed the textual semantics of the provided instructions and selected responses.
2. **Aesthetic Evaluation**: Based on textual aesthetic factors such as paragraph structure, indentation, headings, and subheadings, GPT-4o conducted a detailed textual aesthetic analysis.
3. **Binary Classification**: GPT-4o then performed a binary classification to determine whether the response required modification to improve readability and comprehension.
4. **Revision Process**: For responses that required modification, GPT-4o generated a revised version that preserved the original style and format while enhancing readability and comprehensibility. The prompts requiring aesthetic modifications are documented in the Appendix E.1, which includes the text before and after modification, demonstrating the enhanced readability and comprehension of the revised text.

Finally, we compiled the textual aesthetic preference dataset $\mathcal{D} = \left\{ \left( x^{(i)}, y_t^{(i)}, y_w^{(i)}, y_l^{(i)} \right) \right\}_{i=1}^N$, where $y_t^{(i)}$ represents the revised textual aesthetic data, and $y_w^{(i)}$ and $y_l^{(i)}$ represent the originally chosen and rejected data in UltraFeedback, respectively.

We observed that some polished responses became overly verbose and less natural or human-like. We hypothesize that this is because the original responses in UltraFeedback are already of high quality, making the task of polishing more challenging than expected. To address this issue, we implemented a length constraint for the polishing process. Future work will focus on further improving the textual aesthetic polishing method.

### 3.3 TEXTUAL AESTHETICS SCORING

To validate the aesthetic quality of texts generated by large language models and to assess the effectiveness of our aesthetic preference dataset, a robust method for evaluating text aesthetics is indispensable. Previous studies, such as AlpacaEval (Li et al., 2023; Dubois et al., 2024), MT-Bench (Zheng et al., 2023), and Arena-Hard (Li et al., 2024), suggest that using LLMs as evaluators can effectively approximate human preferences. Consequently, we employ the "LLM as a judge" framework to approximate human preferences for text aesthetics. We evaluate the aesthetic quality of texts generated by LLMs using two methods: text-based and image-based text aesthetic scoring.

**Text-Based Text Aesthetic Scoring.** We randomly selected 500 prompts from Arena-Hard (Li et al., 2024) as our evaluation dataset. Following practices from Arena-Hard and MT-Bench (Zheng et al., 2023), we implemented a pairwise comparison method, comparing the performance of model $\pi_i$ on prompt $p$ with a robust baseline model (GPT-4-0314) to derive aesthetic preference scores. Judges assessed aesthetic preferences on a Likert scale (Likert, 1932) (1 = prefers $\pi_i(p)$ much less than $\pi_{\text{base}}(p)$, 5 = prefers $\pi_i(p)$ much more than $\pi_{\text{base}}(p)$). This methodology ensures that models are penalized more heavily for substantial losses than for minor ones, effectively differentiating between models. Using the chain-of-thought approach, judges evaluated text aesthetics based on four dimensions: readability, visual organization, consistency, and overall structure. To mitigate position bias, we employed a two-game setup by swapping model positions for each query. Following the practices of Chatbot Arena, we adopted the Bradley-Terry (Bradley & Terry, 1952) model to generate final scores. We aggregated all pairwise comparisons with the baseline model and employed bootstrapping to derive a bootstrapped confidence interval for all models' win rates against the baseline, producing an ordered ranking of all models based on their win rates. The judge prompts are provided in Appendix E.2.

**Image-Based Text Aesthetic Scoring.** Our conceptualization of text aesthetics encompasses not only textual readability and comprehensibility but also visual appeal. Given GPT-4o's exceptional multimodal capabilities, we utilized GPT-4o to evaluate text aesthetics from a visual perspective as well. In our experiments, we rendered the LLM-generated texts as HTML with consistent CSS styles, converted them into images of identical size, and then had GPT-4o evaluate these images based on the same criteria used for textual evaluation. Specific prompts are provided in Appendix E.3.

## 4 TEXTUAL AESTHETICS-POWERED TRAINING

### 4.1 DIRECT PREFERENCE OPTIMIZATION TRAINING

Reinforcement Learning with Human Feedback (RLHF)(Christiano et al., 2017) has emerged as a pivotal technique in aligning LLMs (Bai et al., 2022; Ouyang et al., 2022; Stiennon et al., 2020). Early implementations of RLHF primarily relied on reinforcement learning and alternative approaches (Snell et al., 2022; Touvron et al., 2023a; Gulcehre et al., 2023). Rafailov et al. (2024a) proposed a RL-free closedform counterpart known as Direct Preference Optimization (DPO) which has shown impressive performances (Ivison et al., 2023; Jiang et al., 2023; Tunstall et al., 2023).

The naive DPO uses a pair of preference data, which includes a chosen response and a rejected response for each prompt, based on the Bradley-Terry (Bradley & Terry, 1952) model for optimization. The loss function for DPO is defined as follows:

$$\mathcal{L}_{\text{DPO}}(\pi_\theta; \pi_{\text{ref}}) = -\mathbb{E}_{(x,y_w,y_l)\sim\mathcal{D}} \left[ \log \sigma \left( \beta \log \frac{\pi_\theta(y_w \mid x)}{\pi_{\text{ref}}(y_w \mid x)} - \beta \log \frac{\pi_\theta(y_l \mid x)}{\pi_{\text{ref}}(y_l \mid x)} \right) \right], \quad (1)$$

where $\pi_\theta$ denotes the policy being optimized, $\pi_{\text{ref}}$ represents the reference policy, $x$ is the input prompt, $y_w$ is the chosen (winning) response, $y_l$ is the rejected (losing) response, $\mathcal{D}$ is the dataset of prompts and responses, $\sigma$ is the sigmoid function, and $\beta$ is a scaling parameter. By directly integrating preference data into the optimization process, DPO ensures that the generated text aligns closely with human judgments.

## 4.2 TEXTUAL AESTHETICS PREFERENCE OPTIMIZATION TRAINING

For each prompt in our TEXAES dataset, there are three responses: $y_t$, $y_w$, and $y_l$. The response $y_t$ has the same semantic content as $y_w$ but is superior in terms of textual aesthetics. The response $y_w$, in turn, is more aligned with human preferences for chatbots in terms of instruction-following, truthfulness, honesty, and helpfulness compared to $y_l$. The response $y_l$ is the least preferred response in terms of both textual aesthetics and human preferences. The goal of our training is to learn a model that can generate responses that are both aesthetically pleasing and preferred by humans. To achieve this, we designed a textual aesthetics preference optimization (TAPO) approach that jointly optimizes for both textual aesthetics and human preferences.

To simultaneously utilize all three preference data types in the TEXAES dataset for optimization, we adopt the Plackett-Luce (Luce, 1959; Plackett, 1975) model as the underlying preference model. Rafailov et al. (2024a) showed that $\beta \log \frac{\pi_\theta(y|x)}{\pi_{\text{ref}}(y|x)}$ can be treated as "implicit reward" which is assumed to represent the preference for the model generate $y$ given the prompt $x$, the goal of DPO is to align the "implicit reward" towards human preference data directly. We denote each reward function $\beta \log \frac{\pi_\theta(y_k|x)}{\pi_{\text{ref}}(y_k|x)}$ as $r_\theta(x, y_k)$ (where $k \in \{t, w, l\}$), representing the preferences for the model generating $y_t$, $y_w$ $y_l$ given the input $x$. $\pi_\theta$ and $\pi_{\text{ref}}$ are the policy model and reference model respectively and $\beta$ is a hyper-parameter to control the KL divergence between $\pi_\theta$ and $\pi_{\text{ref}}$. The training objective of TAPO is

$$\mathcal{L}_{\text{TAPO}}(\pi_\theta; \pi_{\text{ref}}) =$$
$$- \mathbb{E}_{(x,y_t,y_w,y_l)\sim\mathcal{D}} \left[ \log \left( \frac{\exp(r_\theta(x, y_t))}{\sum_{i\in\{t,w,l\}} \exp(r_\theta(x, y_i))} \cdot \frac{\exp(r_\theta(x, y_w))}{\sum_{i\in\{w,l\}} \exp(r_\theta(x, y_i))} \right) \right] \quad (2)$$

where $\mathcal{D}$ is the dataset, and $\beta$ is the temperature parameter.

Using the properties of logarithmic functions, the loss function can be decomposed into two parts: $\mathcal{L}_{\text{TA}}$ and $\mathcal{L}_{\text{DPO}}$:

$$\mathcal{L}_{\text{TA}} = -\log \left( \frac{\exp(r_\theta(x, y_t))}{\sum_{i\in\{t,w,l\}} \exp(r_\theta(x, y_i))} \right), \quad \mathcal{L}_{\text{DPO}} = -\log \left( \frac{\exp(r_\theta(x, y_w))}{\sum_{i\in\{w,l\}} \exp(r_\theta(x, y_i))} \right). \quad (3)$$

It can be observed that $\mathcal{L}_{\text{DPO}}$ is identical to the loss used in Bradley-Terry model-based preference optimization with $y_w$ and $y_l$, as demonstrated in the proof provided in Appendix C. On the other hand, $\mathcal{L}_{\text{TA}}$ represents the log probability of $r_\theta(x, y_t)$ being ranked first among $r_\theta(x, y_t)$, $r_\theta(x, y_w)$, and $r_\theta(x, y_l)$. $\mathcal{L}_{\text{DPO}}$ primarily optimizes the model's preference for honest, helpful, and truthful data, whereas $\mathcal{L}_{\text{TA}}$ optimizes both the correctness of the answers and textual aesthetics. To ensure the generated answers are not only accurate but also aesthetically pleasing, we assign different weights to the losses to adjust the preference optimization direction. The modified loss function is as follows:

$$\mathcal{L}_{\text{TAPO}}(\pi_\theta, \pi_{\text{ref}}) = -\mathbb{E}_{(x,y_t,y_w,y_l)\sim\mathcal{D}} \left[ w_{\text{TA}} \cdot \mathcal{L}_{\text{TA}} + w_{\text{DPO}} \cdot \mathcal{L}_{\text{DPO}} \right]. \quad (4)$$

## 5 DATA AND EXPERIMENT SETTINGS

### 5.1 TEXTUAL AESTHETICS DATASET

As introduced in Section 3.2, we constructed our textual aesthetic dataset based on a filtered version of UltraFeedback[1] (Cui et al., 2024; Ivison et al., 2023; Tunstall et al., 2023) dataset, which comprises 61,135 completions evaluated by GPT-4, including both accepted and rejected entries. In our experiment, we utilized GPT-4o to perform

| Dataset | #Prompts | Response Length |
|---|---|---|
| ULTRAFEEDBACK | 61,135 | 297 |
| TEXAES | 50,390 | 293 |

Table 1: Statistics of TEXAES datasets.

aesthetic polishing on the UltraFeedback dataset. After the aesthetic polishing process, we found that 5,858 entries were already aesthetically satisfactory and required no further modification. We then analyzed the length of the filtered texts and discovered that a minor subset exhibited excessive verbosity and lacked a natural, human-like quality. To address this, we excluded outliers in the distribution of length differences before and after aesthetic polishing, retaining only data within the 90% confidence interval. We present the statistics of TEXAES in Table 1. We present the length distribution in Appendix A and length constraint filter experiment in Appendix D.

## 5.2 EXPERIMENT SETTINGS

In this study, we evaluate the performance of models from two perspectives: textual aesthetics and general response capabilities. For textual aesthetics, we compare the models using both text-based and image-based text aesthetic scoring methods, as described in Section 3.3. We report the win rate (WR) in text aesthetics at both the text and image levels relative to the baseline model (GPT-4-0314). In addition to automatic evaluation, we conduct a human evaluation to further validate the models' performance. We randomly sample fifty entries from the Anera-Hard dataset and ask human annotators to rate the aesthetics of these entries.

To evaluate the changes in the model's general capabilities following the alignment of textual aesthetics preferences, including its ability to follow instructions and respond to complex prompts across diverse domains, we utilize three well-established auto-evaluation instruction-following benchmarks based on GPT-4-as-a-Judge: AlpacaEval 2.0 (Dubois et al., 2024), Arena-Hard (Li et al., 2024) and MT-Bench (Zheng et al., 2023). For both the supervised tine-tuning and TAPO stages, we employ a low-rank adaptation (Hu et al., 2021) adapter instead of fine-tuning the entire model. Detailed training parameters are provided in the Appendix B.

## 6 EXPERIMENT RESULTS

### 6.1 MAIN RESULTS

The comparative analysis of our models trained with TAPO on TEXAES against open-source models is shown in Table 2. Our LLaMA-3.1-70B-TAPO model surpasses all open-source counterparts in both text-based and image-based text aesthetic metrics, with an 18.88% improvement in text-based scores and a 27.85% enhancement in image-based scores over the best-performing LLaMA-3.1-70B-Instruct model.

For general response benchmarks, the LLaMA-3.1-8B-Instruct and LLaMA-3.1-70B-Instruct models, after TAPO training, show improvements on AlpacaEval 2.0 and MT-Bench, though with a slight decline on Arena-Hard. AlpacaEval 2.0 focuses on chat scenarios, MT-Bench on multi-turn conversations, and Arena-Hard on more complex queries. The gains in AlpacaEval 2.0 and MT-Bench suggest that enhanced text aesthetics contribute to better conversational abilities, aligning with our goal of improving answer clarity, layout, uniformity, and coherence. This underscores the quality of TEXAES and the effectiveness of TAPO in boosting both text aesthetics and overall model performance. Furthermore, the results from experiments using TEXAES and TAPO on Qwen2 (qwe, 2024) and Mistral (Jiang et al., 2023) demonstrate similar performance improvements, showcasing the generalizability of TAPO across diverse model architectures, as detailed in Appendix H.

The results of the human evaluation, shown in Figure 2, show that our LLaMA-3.1-70B-TAPO model is rated significantly higher in text aesthetics than the best-performing open-source model. These results confirm that our model is more visually appealing and coherent, consistent with our quantitative analysis, further validating the efficacy of TAPO in enhancing text aesthetics and overall performance.

---

[1]https://huggingface.co/datasets/HuggingFaceH4/ultrafeedback_binarized

Table 2: Performance comparison between TAPO models and open-source models across various benchmarks."TA Text" and "TA Image" denote text-based and image-based textual aesthetic metrics, respectively. Metrics include: win rates (WR) against GPT-4-Turbo for TA Text and TA Image, WR against GPT-4-0314 for Arena-Hard, length-controlled (LC) win rate against GPT-4-Turbo in AlpacaEval 2.0, and average scores for MT-Bench. All evaluations are conducted using GPT-4 as the judge.

| Model | Size | TA Text WR(%) | TA Image WR(%) | AlpacaEval 2.0 LC WR(%) | Arena-Hard WR(%) | MT-Bench Avg. Score |
|---|---|---|---|---|---|---|
| Qwen2-7B-Instruct (qwe, 2024) | 7B | 24.63 | 39.40 | 33.43 | 27.69 | 7.48 |
| Yi-1.5-9B-Chat (AI et al., 2024) | 9B | 35.52 | 55.03 | 34.74 | **38.89** | 7.38 |
| LLaMA-3.1-8B-Instruct (Dubey et al., 2024) | 8B | 33.42 | 47.94 | 41.34 | 37.10 | 7.42 |
| LLaMA-3.1-8B-TAPO | 8B | **50.85** | **71.91** | **49.84** | 33.89 | **7.72** |
| Tulu-2-dpo-70B (Ivison et al., 2023) | 70B | 9.43 | 27.79 | 31.01 | 16.37 | 6.89 |
| Qwen2-72B-Instruct (qwe, 2024) | 72B | 22.05 | 30.68 | 40.61 | 42.48 | 8.22 |
| LLaMA-3.1-70B-Instruct (Dubey et al., 2024) | 70B | 53.18 | 57.34 | 45.03 | **67.22** | 8.16 |
| LLaMA-3.1-70B-TAPO | 70B | **63.22** | **73.31** | **51.26** | 63.42 | **8.30** |

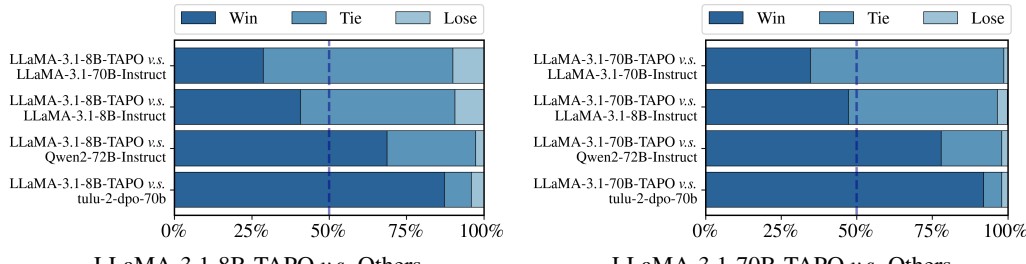

LLaMA-3.1-8B-TAPO *v.s.* Others          LLaMA-3.1-70B-TAPO *v.s.* Others

Figure 2: Win rates of models fine-tuned by TAPO compared to other SOTA open-source models by human judgements in textual aesthetics level. Human judgments are majority votes from three annotators.

## 6.2 IMPACT OF LOSS WEIGHT

To determine the influence of the weight ratio between $\mathcal{L}_{\text{TA}}$ and $\mathcal{L}_{\text{DPO}}$ in TAPO on the aesthetics of the text of the model and the overall performance, we performed a series of methodical experiments. Specifically, we experimented with two settings: 1. First, we used the Tulu-v2 dataset (Ivison et al., 2023) to fine-tune the *LLaMA-3.1-8B-base* model in a supervised manner, followed by further optimization using TAPO; 2. Second, we directly applied TAPO to the *LLaMA-3.1-8B-instruct* model. We set the weight ratios of $\mathcal{L}_{\text{TA}}$ to $\mathcal{L}_{\text{DPO}}$ at 2:1, 1:1, 1:2 and 1:5, respectively, to train the models. We then evaluated the models' text-based and image-based text aesthetic scores, as well as their performance on Arena-Hard.

Figures 3a and 3b illustrate the performance variations of TAPO across different weight ratios. For the *LLaMA-3.1-8B-base* model, increasing the proportion of $\mathcal{L}_{\text{DPO}}$ consistently improves the Arena-Hard score but decreases both text-based and image-based text aesthetic scores. This indicates that a higher proportion of $\mathcal{L}_{\text{DPO}}$ improves optimization toward human preference at the expense of aesthetic preference. For the *LLaMA-3.1-8B-instruct* model, which is already aligned with human preferences, further increasing $\mathcal{L}_{\text{DPO}}$ yields limited improvements in instruct-following capability and significantly decreases textual aesthetic preference.

## 6.3 TWO-STAGE TRAINING

To validate the efficacy of incorporating three types of preference data in TAPO, we conducted a two-stage DPO training ablation experiment. Initially, human preferences were aligned using the $y_w$ and $y_l$ data sets, denoted as DPO($y_w, y_l$). Subsequently, text aesthetic preference alignment was conducted using two methods: DPO($y_t, y_w$) and DPO($y_t, y_l$). These experiments were performed on the *LLaMA-3.1-Base* and *LLaMA3.1-Instruct* models, with results presented in Table 3.

Comparing the final models from the two-stage training with our model trained in TAPO method, we found that, except for the image-based text aesthetic metric, where our model was slightly inferior, it significantly outperformed the two-stage models on text-based aesthetic metrics, AlpacaEval

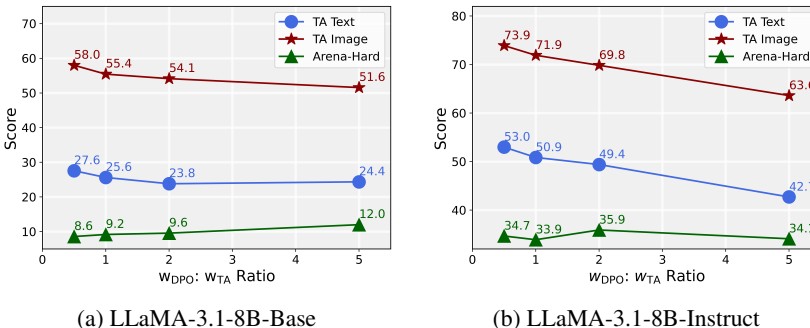

(a) LLaMA-3.1-8B-Base  (b) LLaMA-3.1-8B-Instruct

Figure 3: Performance Across Various Weight Ratios

Table 3: Comparison of two-stage DPO training and TAPO training. $DPO(y_w, y_l)$ + $DPO(y_t, y_l)$ denotes two-stage training where the first stage is $DPO(y_w, y_l)$ and the second stage is $DPO(y_t, y_l)$. The LLaMA-3.1-8B-Base is fine-tuned using the Tulu-v2 dataset.

| Training Settings | TA Text WR(%) | TA Image WR(%) | AlpacaEval 2.0 LC WR(%) | Arena-Hard WR(%) | MT-Bench Avg. Score | MMLU 5-shot |
|---|---|---|---|---|---|---|
| **LLaMA-3.1-8B-Base** | | | | | | |
| $DPO(y_w, y_l)$ + $DPO(y_t, y_l)$ | 25.45 | **60.53** | 23.77 | 7.72 | 5.98 | 63.52 |
| $DPO(y_w, y_l)$ + $DPO(y_t, y_w)$ | 14.03 | 48.31 | 14.66 | 5.35 | 5.50 | 62.80 |
| $TAPO(y_t, y_w, y_l)$ | **25.61** | 55.43 | **26.05** | **9.16** | **6.05** | **64.48** |
| **LLaMA-3.1-8B-Instruct** | | | | | | |
| $DPO(y_w, y_l)$ + $DPO(y_t, y_l)$ | 50.26 | 71.33 | 46.47 | 31.08 | **7.75** | 68.41 |
| $DPO(y_w, y_l)$ + $DPO(y_t, y_w)$ | 50.76 | **75.69** | 44.91 | 29.41 | 7.39 | 67.89 |
| $TAPO(y_t, y_w, y_l)$ | **50.85** | 71.91 | **49.84** | **33.89** | 7.72 | **68.80** |

2.0, Arena-Hard, MT-Bench, and MMLU(Hendrycks et al., 2020). This suggests that TAPO, by leveraging three types of preference data, not only enhances text aesthetic scores but also improves general capabilities.

## 6.4 TEXAES VS. ULTRAFEEDBACK

To validate the effectiveness of the TEXAES data set, we performed a comparative analysis of models trained using TEXAES against those trained with UltraFeedback data. We applied the Direct Preference Optimization (DPO) method to align human preferences with the $y_w$ and $y_l$ pairs from UltraFeedback and the $y_t$ and $y_l$ pairs from TEXAES. The experiments were conducted on both the *LLaMA-3.1-Base* and *LLaMA3.1-Instruct* models.

Table 4: Comparative analysis of TEXAES and UltraFeedback with DPO Training. The baseline represents the performance of LLaMA-3.1-8B-Base which is fine-tuned using the Tulu-v2 dataset and LLaMA-3.1-8B-Instruct.

| Dataset | TA Text WR(%) | TA Image WR(%) | AlpacaEval 2.0 LC WR(%) | Arena-Hard WR(%) | MT-Bench Avg. Score | MMLU 5-shot |
|---|---|---|---|---|---|---|
| **LLaMA-3.1-8B-Base** | | | | | | |
| Baseline | 1.17 | 8.60 | 5.24 | 4.10 | 5.60 | 64.07 |
| UltraFeedback | 2.56 | 8.17 | 9.29 | 7.06 | **5.92** | **65.02** |
| TEXAES | **25.79** | **60.64** | **24.06** | **9.04** | 5.78 | 63.17 |
| **LLaMA-3.1-8B-Instruct** | | | | | | |
| Baseline | 33.42 | 47.94 | 41.34 | **37.10** | 7.42 | 68.80 |
| UltraFeedback | 30.92 | 48.57 | 44.19 | 34.74 | **7.76** | **68.90** |
| TEXAES | **49.07** | **68.63** | **45.82** | 29.87 | 7.55 | 68.52 |

The results, shown in Table 4, indicate that for the *LLaMA-3.1-Base* model, UltraFeedback improved performance in AlpacaEval 2.0, Arena-Hard, MT-Bench and MMLU. For the *LLaMA3.1-Instruct* model, there were performance improvements across most tasks, except for a slight decline in Arena-Hard. However, UltraFeedback did not improve performance in aesthetic evaluation tasks. Models trained with TEXAES showed significant performance improvements over those trained with UltraFeedback in most tasks on the *LLaMA-3.1-Base* model, with a minor decrease in MMLU. For the *LLaMA3.1-Instruct* model, the one trained with TEXAES exhibited general capabilities comparable

to those of the UltraFeedback-trained model while surpassing it in aesthetic tasks. These experiments demonstrate that TEXAES not only optimizes the textual aesthetic performance of large language models but also aligns well with human preferences.

## 6.5 ANNOTATION CONSISTENCY

We generated responses for 50 questions sampled from Arena-Hard using six models: *LLaMA-3.1-8B-TAPO*, *LLaMA-3.1-70B-TAPO*, *LLaMA-3.1-8B-Instruct* (Dubey et al., 2024), *LLaMA-3.1-70B-Instruct* (Dubey et al., 2024), *Qwen2-72B-Instruct* (qwe, 2024), and *Tulu-2-dpo-70B* (Ivison et al., 2023). Subsequently, we employed three types of evaluators: text-based GPT-4o judge (TA Text), image-based GPT-4o judge (TA Image), and three human annotators (details can be found in Appendix G). Each evaluator was tasked with comparing *LLaMA-3.1-8B-TAPO* and *LLaMA-3.1-70B-TAPO* against other models in terms of the textual aesthetics of the generated answers (win/tie/lose), resulting in 400 annotated comparison pairs.

Table 5 presents the agreement ratios, as utilized in MT-Bench (Zheng et al., 2023), among the TA Text scores, TA Image scores, and annotators, as well as annotators themselves. On average, the TA Text scores demonstrated a 68.67% agreement rate with the human annotators, while the TA Image scores judges exhibited a 64.83% agreement rate, which is lower than that of the human annotators. Notably, the agreement rates of both our image-based and text-based GPT-4o judges are comparable to those observed in previous human evaluations, which reported an average of 66% agreement in MT-Bench (Zheng et al., 2023) and 59.7% in UltraFeedback (Cui et al.,

Table 5: Agreement between judges and human annotators on 400 samples from Arena-Hard. A-1, A-2, and A-3 are three human annotators. TA Text is the text-based GPT-4o judge, and TA Image is the image-based GPT-4o judge.

| Judge | A-1 | A-2 | A-3 | Average |
|---|---|---|---|---|
| A-1 | - | 78.25% | 77.50% | 77.88% |
| A-2 | 78.25% | - | 80.75% | 79.50% |
| A-3 | 77.50% | 80.75% | - | 79.13% |
| TA Image | 60.75% | 68.00% | 65.75% | 64.83% |
| TA Text | 69.00% | 70.00% | 67.00% | 68.67% |

2024). These results suggest that our GPT-4o judges can serve as effective proxies for human preferences in assessing text aesthetics.

## 6.6 CRITERIA FOR REJECT SAMPLE SELECTION

To effectively optimize textual aesthetics using preference optimization, it is essential to construct preference pairs consisting of chosen and rejected responses. For our purposes, we select $y_t$ from TEXAES as the chosen response. As the rejected response, we use either the original chosen response $y_w$ or the original rejected response $y_l$ from the UltraFeedback dataset. We conducted DPO experiments to compare the impact of $y_w$ and $y_l$ on the model's performance. The results are presented in Table 6.

Table 6: Evaluation of performance across different rejected samples.

| Training Settings | TA Text WR(%) | TA Image WR(%) | AlpacaEval 2.0 LC WR(%) | Arena-Hard WR(%) | MT-Bench Avg. Score | MMLU 5-shot |
|---|---|---|---|---|---|---|
| **LLaMA-3.1-8B-Base** | | | | | | |
| Baseline | 1.17 | 8.60 | 5.24 | 4.10 | 5.60 | **64.07** |
| DPO$(y_t, y_w)$ | 15.72 | 51.70 | 15.78 | 4.10 | 5.19 | 50.36 |
| DPO$(y_t, y_l)$ | **25.79** | **60.64** | **24.06** | **9.04** | **5.78** | 63.17 |
| **LLaMA-3.1-8B-Instruct** | | | | | | |
| Baseline | 33.42 | 47.94 | 41.34 | **37.10** | 7.42 | **68.80** |
| DPO$(y_t, y_w)$ | 46.89 | **71.19** | 38.93 | 26.04 | 7.36 | 68.31 |
| DPO$(y_t, y_l)$ | **49.07** | 68.63 | **45.82** | 29.87 | **7.55** | 68.52 |

The model trained with DPO$(y_t, y_l)$ outperforms the model trained with DPO$(y_t, y_w)$ in most tasks. This superior performance is attributed to DPO$(y_t, y_l)$ optimizing both textual aesthetic preferences and human preferences. When using $y_w$ as the rejected response, the model trained with DPO$(y_t, y_w)$ significantly outperforms the baseline model in text aesthetic scores but shows a decrease in performance in other tasks. This indicates that $y_t$ is more aligned with aesthetic preferences, meeting our expectations.

## 7 CASE STUDY

In this section, we compare *LLaMA-3.1-8B-Instruct* and *LLaMA-3.1-8B-TAPO* from three cases in Arena-Hard, as shown in Figure 4.

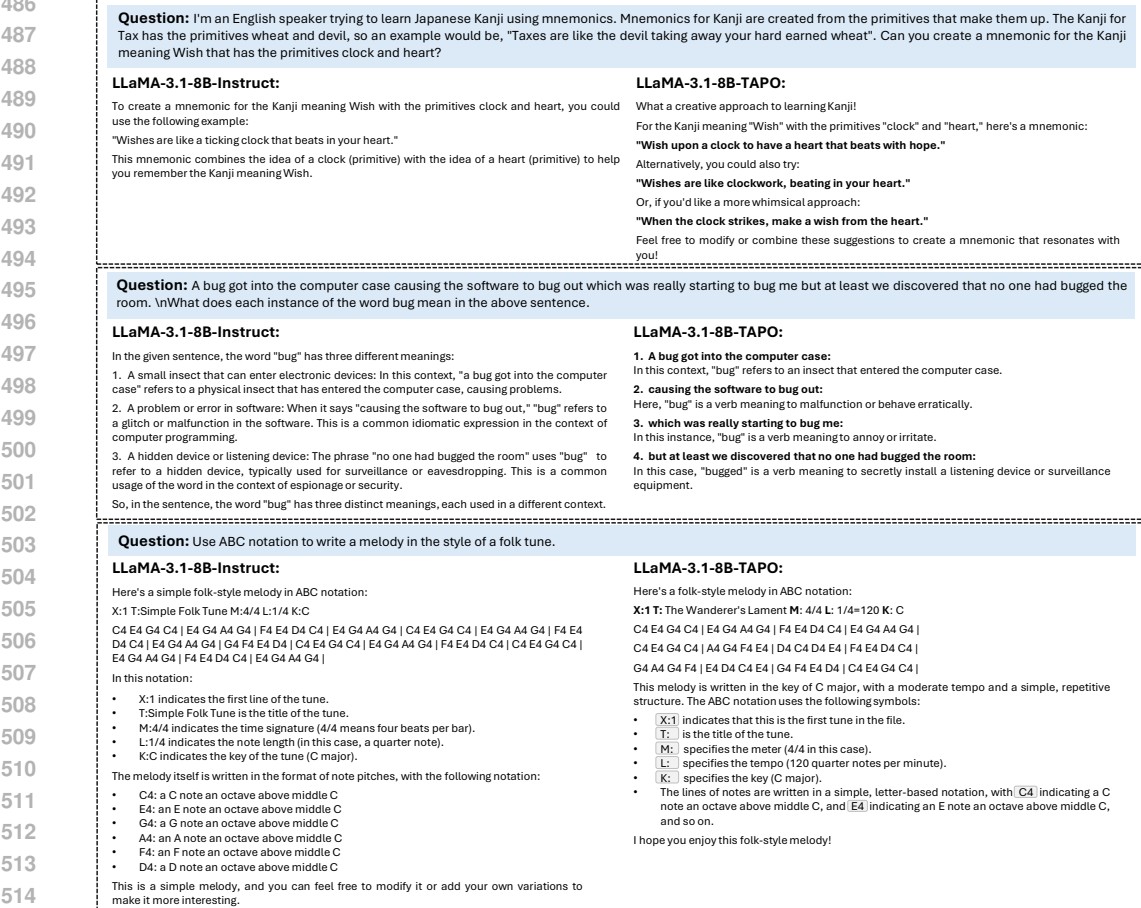

Figure 4: Three cases in Arena-Hard.

The first case (top of Figure 4) demonstrates how *LLaMA-3.1-8B-TAPO* improves the mnemonic for the Kanji character 'Wish' by providing multiple thoughtfully separated options, each clearly formatted and logically structured. This enhances clarity and allows learners to identify and select a mnemonic that resonates with them, compared to the single, less engaging option by *LLaMA-3.1-8B-Instruct*. In the second case (center of Figure 4), *LLaMA-3.1-8B-TAPO* improves readability and comprehension by using bold formatting to emphasize each occurrence of 'bug' and aligning explanations with a numbered list. This ensures better organization and enables readers to quickly grasp the context and meaning of each instance, whereas *LLaMA-3.1-8B-Instruct*'s less structured formatting is harder to follow. In the third case (bottom of Figure 4), *LLaMA-3.1-8B-TAPO* organizes a folk-style melody with logical grouping of notes and appropriate line breaks, enhancing readability and usability. In contrast, the folk-style melody output by *LLaMA-3.1-8B-Instruct* suffers from fragmented line breaks, splitting logical sequences of notes into disjoint segments, which disrupts the logical flow and makes it challenging to interpret and perform the melody accurately.

## 8 CONCLUSION

In this paper, we conducted the first exploration of textual aesthetics in LLMs and introduced a series of techniques to enhance the aesthetic quality of LLMs outputs. First, we developed the TEXAES dataset, the first textual aesthetic dataset in the LLMs domain, using our specially-designed data polishing pipeline. Based on this dataset, we proposed the TAPO, which fine-tunes LLMs to improve the aesthetic quality of their outputs while preserving their core capabilities. Both qualitative and quantitative experiments validated the effectiveness of our proposed techniques. We hope our work serves as an early exploration for the textual aesthetics in LLMs and provides valuable support for researchers in the open-source community. In future work, we will continue to explore ways to collect diverse and high-quality textual aesthetics data, while designing more efficient and effective tuning techniques for aesthetic fine-tuning.

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

## A  DATASET STATISTICS

Figure 5 illustrates the difference in token length between the text that has undergone aesthetic polishing and the original text. The mean length difference is 49 tokens, with the 25th and 75th percentile values being -7 and 54, respectively. The maximum length difference is 2673 tokens, while the minimum length difference is -1024 tokens.

In Figure 6, we plot the length distribution of our TEXAES. The mean length is 293 tokens, with the 25th and 75th percentile values being 97 and 444 respectively, and the maximum length being 1408 tokens. Figure 6 shows the length distribution of UltraFeedback (Cui et al., 2024). The mean length is 297 tokens, with the 25th and 75th percentile values being 77 and 464 respectively, and the maximum length being 2700 tokens.

## B  TRAINING PARAMETERS

We present the details of the experimental settings in Table 7 and Table 8. For the sake of fairness in comparison, we used the same training parameters as those employed by DPO during the preference optimization stage. Our experiments are based on Llama-Factory (Zheng et al., 2024)

## C  MATHEMATICAL DERIVATIONS

In this section, we prove that $\mathcal{L}_{\text{DPO}}$ from Eq. 3 is equivalent to Eq. 1. To begin, consider Eq.3:

$$
\begin{aligned}
\mathcal{L}_{\text{DPO}} &= -\log\left(\frac{\exp(r_\theta(x, y_w))}{\sum_{i\in\{w,l\}} \exp(r_\theta(x, y_i))}\right) \\
&= -\log\left(\frac{\exp(r_\theta(x, y_w))}{\exp(r_\theta(x, y_w)) + \exp(r_\theta(x, y_l))}\right) \\
&= -\log\left(\frac{1}{1 + \exp(r_\theta(x, y_l) - r_\theta(x, y_w))}\right) \\
&= -\log\sigma\left(r_\theta(x, y_w) - r_\theta(x, y_l)\right)
\end{aligned}
\tag{5}
$$

Here, $\sigma$ denotes the sigmoid function. In Section 4.2, we presented the specific expressions for $r_\theta(x, y_w)$ and $r_\theta(x, y_l)$:

$$
r_\theta(x, y_w) = \beta\log\frac{\pi_\theta(y_w \mid x)}{\pi_{\text{ref}}(y_w \mid x)}, \quad r_\theta(x, y_l) = \beta\log\frac{\pi_\theta(y_l \mid x)}{\pi_{\text{ref}}(y_l \mid x)}
\tag{6}
$$

By substituting Eq. 6 into the Eq. 5, we obtain:

$$
\mathcal{L}_{\text{DPO}} = -\log\sigma\left(\beta\log\frac{\pi_\theta(y_w \mid x)}{\pi_{\text{ref}}(y_w \mid x)} - \beta\log\frac{\pi_\theta(y_l \mid x)}{\pi_{\text{ref}}(y_l \mid x)}\right)
\tag{7}
$$

This shows that $\mathcal{L}_{\text{DPO}}$ as defined in Eq. 3 is indeed equivalent to Eq. 1, thus completing the proof.

## D  LENGTH CONSTRAINT IN TEXAES DATASET

To verify whether filtering out outliers in the distribution of length differences before and after aesthetic polishing can improve the quality of TEXAES during its construction phase, we conducted an ablation experiment on data without length filtering. Specifically, the model was trained based on *LLaMA-3.1-8B-Base* using DPO($y_t$, $y_l$), with the outcomes delineated in Table 9. The findings demonstrate that the performance of the model, after removing data points with excessive length deviations, significantly exceeds that of the model trained without such length filtering across all evaluation tasks. Furthermore, a statistical analysis of the output lengths generated by the model revealed that the outputs produced by the model trained with length-filtered data were not only shorter but also more concise, thereby affirming the efficacy of length filtering in text aesthetic optimization.

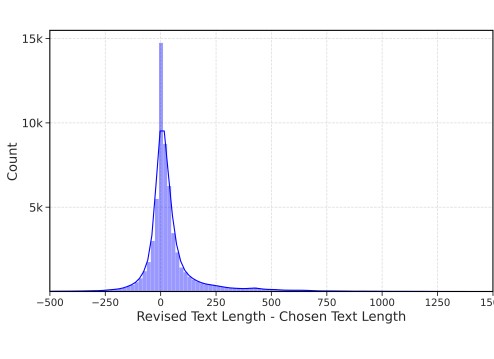

Figure 5: Distribution of length differences.

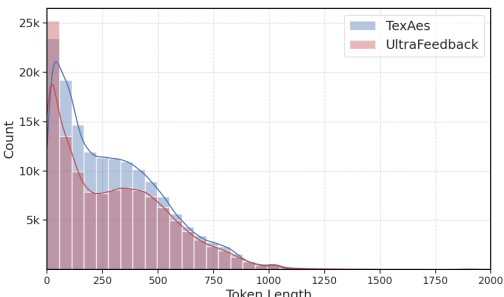

Figure 6: Comparison of token length distributions between TEXAES and UltraFeedback.

Table 7: Parameters for SFT training.

| Parameter | Value |
|---|---|
| Training Method | LoRA (Hu et al., 2021) |
| Maximum Sequence Length | 2048 |
| Optimizer | AdamW |
| Precision | BFloat16 |
| Global Batch Size | 64 |
| Maximum Learning Rate | 0.0002 |
| Learning Rate Scheduler | Cosine with 10% Warmup |
| Number of Epochs | 2 |

Table 8: Parameters for TAPO training.

| Parameter | Value |
|---|---|
| Training Method | LoRA (Hu et al., 2021) |
| Maximum Sequence Length | 2048 |
| Optimizer | AdamW |
| Precision | BFloat16 |
| Global Batch Size | 64 |
| Maximum Learning Rate | 0.00002 |
| Learning Rate Scheduler | Cosine with 10% Warmup |
| Number of Epochs | 2 |
| Beta | 0.1 |
| Loss Weight $w_{\text{TA}}$ | 1 |
| Loss Weight $w_{\text{DPO}}$ | 1 |

Table 9: Ablation study for length filter.

| Length Filter | TA Text | TA Image | AlpacaEval 2.0 | Arena-Hard | MT-Bench | Avg Tokens |
|---|---|---|---|---|---|---|
| ✗ | 24.94 | 56.64 | 20.62 | 7.57 | 4.75 | 649 |
| ✓ | 25.79 | 60.64 | 24.06 | 9.04 | 5.78 | 610 |

# E PROMPT

## E.1 AESTHETICS POLISHING PROMPT

> **Prompt Template for Text Rewriting**
>
> **System Instruction**
> You are tasked with acting as a text rewriter to enhance the readability and comprehension of text generated by a Large Language Model (LLM). Your goal is to ensure the text is easy to read, easy to understand, and visually organized in a logical manner. Modifications should be reasonable and appropriate, rather than mandatory. Each element should be used judiciously to enhance readability and comprehension.
> **User Instruction**
> <|User Prompt|>
> {instruction}
> <|The Start of Assistant's Answer|>
> {completion}
> <|The End of Assistant's Answer|>
>
> Your task is to:
> 1. **Analyze the LLM-generated response**:
>     - Read and understand the text to grasp its context and purpose.
>     - Carefully review the text generated by the LLM.
>     - Evaluate its structure, formatting, and overall readability.
> 2. **Determine the Need for Modification**:
>     - Decide whether the text needs modification to improve its readability and comprehension.
>     - If the text is already satisfactory, no changes are necessary.
> 3. **Provide a Revised Version of the Text if Necessary**:
>     - Make appropriate modifications to enhance the text's readability and comprehension.
>     - Ensure the revised text maintains a consistent style and format throughout.
>
> **Textual Aesthetic Elements to Consider**:
> 1. **Paragraph Structure**: Ensure paragraphs are of appropriate length and logically structured. Use appropriate spacing between paragraphs.
> 2. **Indentation**: Apply consistent indentation if necessary.
> 3. **Headings and Subheadings**: Use headings to organize content and improve readability, but only if the content naturally lends itself to such structure.
> 4. **Lists and Bullet Points**: Utilize lists to break down complex information when applicable.
> 5. **Formatting for Emphasis**: Use bold or italic text to emphasize important points judiciously.
> 6. **Line Spacing**: Adjust line spacing to enhance readability.
> 7. **Consistency**: Maintain a consistent style throughout the document.
> 8. **Visual Breaks**: Use visual breaks to separate different sections if applicable.
> 9. **Blockquotes**: Use blockquotes for quotations or highlighted text.
> 10. **Links**: Format hyperlinks appropriately when applicable.
> 11. **Tables**: Use tables for any tabular data if required.
> 12. **Whitespace and Spacing**: Ensure appropriate use of whitespace and spacing to avoid a cluttered appearance.
>
> **Format**:
> **Textual Aesthetic Analysis**:
> - Your analysis
> **Does it need modification**:
> - If it needs modification: [[Y]]
> - If it doesn't need modification: [[N]]
> **Revised Text**:

- If it needs modification: <|Revised Content Start|>Your revised text<|Revised Content End|>
- If it doesn't need modification: <|Revised Content Start|>""<|Revised Content End|>

**Example Output**:
**Textual Aesthetic Analysis**:
The text is clear, well-organized, and easy to read.
**Does it need modification**: [[N]]
**Revised Text**:
<|Revised Content Start|>""<|Revised Content End|>

## E.2 TEXT-BASED TEXTUAL AESTHETICS SCORING PROMPT

---

**Prompt Template for Text-Based TEXTUAL AESTHETICS Scoring**

**System Instruction**
You are an impartial judge tasked with evaluating the textual aesthetics of responses provided by two AI assistants to the user prompt displayed below. Your goal is to determine which response is more aesthetically pleasing and easier to read and understand.

Begin your evaluation by considering the following aspects for each response:

1. **Readability**: Is the text easy to read and understand? Are the sentences of appropriate length and complexity?
2. **Visual Organization**: Is the text visually organized in a logical manner? Are there appropriate headings, subheadings, lists, and other formatting elements?
3. **Consistency**: Does the text maintain a consistent style and format throughout?
4. **Overall Structure**: Are the paragraphs well-structured and logically connected? Is there appropriate spacing between paragraphs?

Follow these steps for your evaluation:
1. **Analyze each response**: Carefully read and analyze both responses based on the criteria provided.
2. **Compare both responses**: Determine which response excels in textual aesthetics considering all aspects.
3. **Make a final decision**: Choose the response that is better in terms of textual aesthetics and justify your choice.

You must output only one of the following choices as your final verdict with a label:
1. Assistant A is significantly better: [[A>>B]]
2. Assistant A is slightly better: [[A>B]]
3. Tie, relatively the same: [[A=B]]
4. Assistant B is slightly better: [[B>A]]
5. Assistant B is significantly better: [[B>>A]]

Example output: "My final verdict is Assistant A is slightly better: [[A>B]]."
**User Instruction**
<|User Prompt|>
{question}
<|The Start of Assistant A's Answer|>
{answer_1}
<|The End of Assistant A's Answer|>

<|The Start of Assistant B's Answer|>
{answer_2}
<|The End of Assistant B's Answer|>"

---

972
973
974
975
976
977
978
979
980
981
982
983
984
985
986
987
988
989
990
991
992
993
994
995
996
997
998
999
1000
1001
1002
1003
1004
1005
1006
1007
1008
1009
1010
1011
1012
1013
1014
1015
1016
1017
1018
1019
1020
1021
1022
1023
1024
1025

## E.3 IMAGE-BASED TEXTUAL AESTHETICS SCORING PROMPT

---

### Prompt Template for Image-Based TEXTUAL AESTHETICS Scoring

**System Instruction**

You are an impartial judge tasked with evaluating the textual and visual aesthetics of responses provided by two AI assistants to the user prompt displayed below. You will be given both the textual answers and images of the responses from each assistant. Your goal is to determine which response is more aesthetically pleasing and easier to read and understand, considering both textual and visual factors.

Evaluate each response based on the following criteria:

1. **Readability**: Is the text easy to read and understand? Are the sentences of appropriate length and complexity?
2. **Visual Organization**: Is the text visually organized in a logical manner? Are there appropriate headings, subheadings, lists, and other formatting elements?
3. **Consistency**: Does the text maintain a consistent style and format throughout?
4. **Overall Structure**: Are the paragraphs well-structured and logically connected? Is there appropriate spacing between paragraphs?

Follow these steps for your evaluation:
1. **Analyze each response**: Carefully examine both images based on the criteria provided.
2. **Compare both responses**: Determine which response excels in textual and visual aesthetics considering all aspects.
3. **Make a final decision**: Choose the response that is better in terms of textual and visual aesthetics and justify your choice.

Output your final verdict with one of the following labels:
1. Assistant A is significantly better: [[A>>B]]
2. Assistant A is slightly better: [[A>B]]
3. Tie, relatively the same: [[A=B]]
4. Assistant B is slightly better: [[B>A]]
5. Assistant B is significantly better: [[B>>A]]

Example output:
1. Analysis of Assistant A's response:
      - Readability: ...
      - Visual Organization: ...
      - Consistency: ...
      - Overall Structure: ...
2. Analysis of Assistant B's response:
      - Readability: ...
      - Visual Organization: ...
      - Consistency: ...
      - Overall Structure: ...
3. Comparison:
      - Both responses are similar in readability, but...
      - Assistant A has better visual organization...
      - Assistant B's consistency is...
      - Overall, Assistant A/B stands out in...

My final verdict is Assistant A is slightly better: [[A>B]].
**User Instruction**
<|User Prompt|>
{question}

Below are two images: the first one is Assistant A's response, and the second one is Assistant B's response. Please evaluate them based on the criteria provided and give the final verdict answer.
<|The Image of Assistant A's Answer|>
{base64_image1}
<|The Image of Assistant B's Answer|>
{base64_image2}

# F ADDITIONAL EXAMPLES

## F.1 TEXT AESTHETICS JUDGEMENT

---

**Question:** A bug got into the computer case causing the software to bug out which was really starting to bug me but at least we discovered that no one had bugged the room. \nWhat does each instance of the word bug mean in the above sentence.

**Assistant A:**

In the sentence you provided, the word "bug" is used in several different contexts, each with a different meaning:

1. "A bug got into the computer case"
• In this context, "bug" refers to an insect or small creature that has physically entered the inside of the computer case.

2. "causing the software to bug out"
• Here, "bug out" is a colloquial expression meaning that the software started to malfunction or behave erratically. The term "bug" in software refers to an error or flaw in the program that causes it to operate incorrectly.

3. "which was really starting to bug me"
• In this instance, "bug" is used as a verb meaning to annoy or bother the speaker.

4. "no one had bugged the room"
• Finally, in this context, "bugged" means to have secretly placed an electronic listening device (or "bug") in the room for the purpose of surveillance or eavesdropping.

Each use of "bug" demonstrates the versatility of the word in the English language, where it can be a noun or a verb with several different meanings depending on the context.

**Assistant B:**

In the given sentence, the word "bug" has three different meanings:

1. A small insect that can enter electronic devices: In this context, "a bug got into the computer case" refers to a physical insect that has entered the computer case, causing problems.

2. A problem or error in software: When it says "causing the software to bug out," "bug" refers to a glitch or malfunction in the software. This is a common idiomatic expression in the context of computer programming.

3. A hidden device or listening device: The phrase "no one had bugged the room" uses "bug" to refer to a hidden device, typically used for surveillance or eavesdropping. This is a common usage of the word in the context of espionage or security.

So, in the sentence, the word "bug" has three distinct meanings, each used in a different context.

**GPT-4o Judgement:**

1. Analysis of Assistant A's response:
 - **Readability**: The text is clear and easy to read. Each instance of "bug" is explained in a straightforward manner.
 - **Visual Organization**: The response is well-organized with numbered points, making it easy to follow. The use of bullet points helps in distinguishing each meaning.
 - **Consistency**: The style and format are consistent throughout the response.
 - **Overall Structure**: The paragraphs are well-structured, and there is appropriate spacing between points. The concluding sentence ties the explanation together nicely.
2. Analysis of Assistant B's response:
 - **Readability**: The text is also clear and easy to read. The explanations are concise and to the point.
 - **Visual Organization**: The response is organized with numbered points, but it lacks the additional detail and separation seen in Assistant A's response.
 - **Consistency**: The style and format are consistent throughout the response.
 - **Overall Structure**: The paragraphs are well-structured, but the response is slightly less detailed compared to Assistant A's. The concluding sentence is effective but less comprehensive.

3. Comparison:
 - Both responses are similar in readability, but Assistant A provides more detailed explanations.
 - Assistant A has better visual organization with more spacing and separation between points.
 - Both responses maintain consistency in style and format.
 - Overall, Assistant A stands out due to its more detailed and visually organized response.

My final verdict is Assistant A is slightly better: [[A>B]].

---

Figure 7: An example of image-based GPT-4o judgement. The Assistant A is GPT-4 Turbo, and Assistant B is LLaMA-3.1-8B-Instruct. The image demonstrates that GPT-4o can evaluate text aesthetics, showing that Assistant A's response is more visually organized and detailed compared to Assistant B's.

# G HUMAN ANNOTATION DETAILS

In our study, we employed three annotators: two graduate students in computer science and one professor with a background in applied linguistics. All three evaluators are non-native English speakers but are proficient in English. Their diverse academic and linguistic backgrounds provide a balanced perspective for assessing textual aesthetics across the four key dimensions—clarity, layout, uniformity, and coherence.

The annotators underwent a comprehensive training and calibration process prior to the main evaluation. This training ensured that their understanding of the evaluation criteria was consistent and aligned. Annotators were introduced to the four evaluation dimensions—clarity (ease of comprehension), layout (visual organization), uniformity (consistent formatting), and coherence (logical structure)—with detailed explanations and examples. They practiced with a subset of the dataset, and their evaluations were reviewed with feedback provided to refine their approach. A final readiness test was conducted to confirm alignment and preparedness for the main evaluation phase.

For the evaluation, 50 prompts were randomly selected from the Arena-Hard (Li et al., 2024), and all models under evaluation were tasked with generating responses to these prompts using identical parameters. This ensured consistency in the generation process and a fair basis for comparison across models. The generated text samples were anonymized and presented in a standardized format, removing all identifying information about the originating model or source.

The annotators independently evaluated these samples without communication or influence from others, maintaining impartiality throughout the process. A pairwise comparison methodology was employed, where annotators assigned scores in the form of win, tie, or loss for each sample comparison across the four evaluation dimensions.

## H GENERALIZABILITY OF TEXAES AND TAPO TO OTHER LLMS

To evaluate the generalizability of the proposed TEXAES dataset and the TAPO method beyond the LLaMA series, we conducted additional experiments on two other widely used large language models: Qwen2-7B-Instruct (qwe, 2024) and Mistral-7B-Instruct-v0.3 (Jiang et al., 2023).

In these experiments, TEXAES was used as the training dataset, and TAPO was applied as the training method under the same experimental settings as those used for LLaMA-3.1-8B-Instruct. The evaluation results, summarized in Table 10, show significant improvements in textual aesthetics for both Qwen2-7B-Instruct and Mistral-7B-Instruct-v0.3 after the application of TAPO. Specifically, both models demonstrated significant enhancements in textual aesthetics and general response capabilities following training with TAPO.

These findings are consistent with the results observed in the LLaMA-3.1 series, providing compelling evidence of the broad applicability and effectiveness of the TEXAES dataset and the TAPO method across diverse LLM architectures.

Table 10: Performance comparison of Qwen2-7B-Instruct and Mistral-7B-Instruct-v0.3 models after training with TEXAES and TAPO

| Model | TA Text WR(%) | TA Image WR(%) | AlpacaEval 2.0 LC WR(%) | Arena-Hard WR(%) | MT-Bench Avg. Score | MMLU 5-shot |
|---|---|---|---|---|---|---|
| **Qwen2-7B-Instruct** | | | | | | |
| Qwen2-7B-Instruct (qwe, 2024) | 24.63 | 39.40 | 33.43 | 27.69 | 7.48 | 70.46 |
| Qwen2-7B-Instruct + DPO $(y_t, y_l)$ | 33.84 | 61.23 | 40.16 | 25.30 | 7.19 | 70.34 |
| Qwen2-7B-Instruct-TAPO | **37.99** | **64.28** | **40.27** | **32.40** | **7.48** | **70.49** |
| **Mistral-7B-Instruct-v0.3** | | | | | | |
| Mistral-7B-Instruct-v0.3 (Jiang et al., 2023) | 8.26 | 28.90 | 29.87 | 17.13 | 6.59 | 61.52 |
| Mistral-7B-Instruct-v0.3 + DPO $(y_t, y_l)$ | 25.59 | 54.64 | 36.78 | 20.83 | 6.56 | 61.36 |
| Mistral-7B-Instruct-v0.3-TAPO | **28.55** | **57.84** | **38.53** | **23.10** | **6.80** | **61.55** |

