# OpenReview forum: "Textual Aesthetics in Large Language Models"
_ICLR.cc/2025/Conference — ICLR 2025 Conference Withdrawn Submission_

### Official Review · Reviewer_NtqG · 2024-10-30

**Soundness:** 2
**Presentation:** 3
**Contribution:** 2
**Rating:** 5
**Confidence:** 4

**Summary:**

The paper pioneers the exploration of textual aesthetics in Large Language Models (LLMs). The authors propose an aesthetic refinement process and create a dataset named TEXAES. Based on this dataset, they introduce a textual aesthetics optimization fine-tuning method called TAPO, which aims to enhance textual aesthetics without compromising content accuracy. Additionally, the paper develops two textual aesthetics assessment methods, one based on text analysis and the other on image analysis, to evaluate the aesthetics of LLM outputs. Experimental results indicate that using the TEXAES dataset and TAPO fine-tuning method not only improves aesthetic scores but also enhances model performance on general evaluation datasets such as AlpacalEval and Anera-hard.

**Strengths:**

1. The paper is the first to investigate textual aesthetics in LLMs, introducing the TEXAES dataset and TAPO fine-tuning method, providing a new direction for the aesthetic optimization of LLMs.
2. The paper empirically validates the effectiveness of the TEXAES dataset and TAPO method, demonstrating not only improved aesthetic scores but also enhanced model performance.
3. The paper develops two assessment methods based on text and image analysis, offering tools for a comprehensive evaluation of the aesthetics of LLM outputs.

**Weaknesses:**

1. The paper does not elaborate on the quantitative standards for textual aesthetics, that is, what constitutes good textual aesthetics, and how the consistency among human evaluators in textual aesthetics assessment is ensured.
2. From the examples shown in Figure 4, the paper's concept of textual aesthetics seems to involve only line breaks, bold fonts, and highlighting key points, which are relatively simple and may lack long-term research value.
3. The paper does not clearly explain how to determine if texts with better textual aesthetics are better understood, and how to ensure that human evaluators are making judgments based on content comprehension rather than just neat formatting.
4. The paper uses GPT-4o for automated assessment, but as a machine, GPT-4o processes only code/symbols/characteristics, and its suitability for evaluating textual aesthetics is questionable.

**Questions:**

See Weaknesses.

---

> ### Author Response · Authors · 2024-11-18
> **Rebuttal by Authors (1/4)**
>
> We are grateful to the Reviewer for the extensive review. We address your questions point by point below.
>
> > **Q1: The paper does not elaborate on the quantitative standards for textual aesthetics, that is, what constitutes good textual aesthetics, and how the consistency among human evaluators in textual aesthetics assessment is ensured.**
>
> **A1**: As described in Section 3.3, textual aesthetics are evaluated across four fundamental dimensions: clarity (ease of comprehension), layout (visual organization), uniformity (consistent formatting), and coherence (logical structure). These dimensions are assessed together, resulting in a comprehensive Likert-scale evaluation (1–5) during pairwise comparisons against a baseline model (GPT-4-0314). For robustness, the Bradley-Terry model is utilized to aggregate these pairwise comparisons into a unified ranking, ensuring consistency and quantitative evaluation of textual aesthetics.
>
> To ensure consistency among human evaluators, we implemented a rigorous preparation process. Before starting the annotation task, evaluators underwent a learning phase to familiarize themselves with the definitions and evaluation criteria for textual aesthetics. They were instructed to comprehensively analyze and compare text samples across the four dimensions (clarity, layout, uniformity, and coherence) to assign scores (win/tie/loss) for textual aesthetics between models. Each annotator was tested on a subset of the dataset to confirm their understanding of these definitions prior to commencing the full annotation task.
>
> Additionally, consistency among evaluators was verified through an inter-annotator agreement study (reported in Section 6.5, Table 5), which showed an average agreement rate of 78.84% among annotators which is comparable to those observed in previous human evaluations consistency(MT-Bench[1], Ultrafeedback[2]), confirming that evaluators consistently applied the evaluation criteria. This process ensures reliable and consistent textual aesthetics assessment.
>
> ---
>
> **Reference**
>
> [1] Zheng, Lianmin, et al. "Judging llm-as-a-judge with mt-bench and chatbot arena." NeurIPS 2023.
>
> [2] Cui, Ganqu, et al. "Ultrafeedback: Boosting language models with scaled ai feedback" ICML 2024.

---

> ### Author Response · Authors · 2024-11-18
> **Rebuttal by Authors (2/4)**
>
> > **Q2: From the examples shown in Figure 4, the paper's concept of textual aesthetics seems to involve only line breaks, bold fonts, and highlighting key points, which are relatively simple and may lack long-term research value.**
>
> **A2:** Thank you for your observation. To address your concerns, we will clarify two key aspects: (1) the comprehensive scope of textual aesthetics, and (2) the long-term research potential of textual aesthetics for LLMs.
>
> **1. The Comprehensive Scope of Textual Aesthetics**
>
> We would like to clarify that our concept of textual aesthetics is not limited to elements such as line breaks, bold fonts, and highlighting key points. As described in Section 3.1, our definition encompasses broader and more foundational dimensions, including clarity (ease of comprehension), layout (visual organization), uniformity (consistent formatting), and coherence (logical structure). These dimensions aim to improve not only the visual appeal of the text but also its readability and logical flow, which are critical for effective communication and understanding.
>
> Regarding the specific example in Figure 4, the use of line breaks, bold fonts, and other elements must be appropriate and meaningful. For instance, in the third case (bottom of Figure 4), the folk-style melody output by LLaMA-3.1-8B-Instruct lacks proper line breaks, resulting in fragmented sequences of notes where four-syllable blocks are split. This diminishes readability, visual organization, and coherence, making it harder for users to interpret and utilize the melody.
>
> In contrast, the output from LLaMA-3.1-8B-TAPO uses appropriate line breaks and a neatly arranged structure. This significantly enhances clarity, layout, and logical structure, making the melody easier to read and effectively perform. Such thoughtful formatting not only aligns with the principles of textual aesthetics but also ensures the usability of the text for musical purposes.
>
> **2. Long-Term Research Potential of Textual Aesthetics**
>
> Textual aesthetics plays a foundational role in optimizing the usability and engagement of content generated by LLMs. Drawing parallels with the evolution of **image aesthetics**, we note that visual appeal has significantly influenced the adoption and success of models for image generation, with aesthetic fine-tuning contributing to more human-aligned outputs in fields like design, photography, and multimedia content creation. Similarly, **textual aesthetics** holds the potential to transform how users interact with text-based outputs from LLMs by ensuring that the content is not only accurate but also structured and presented in a way that aligns with human preferences.
>
> Textual aesthetics is a critical area for advancing the quality and utility of LLMs, and its long-term research value spans a range of potential directions, including but not limited to:
>
> 1. **Enhanced User Engagement**: Aesthetic textual outputs increase engagement by making content more readable, visually appealing, and intuitive.
> 2. **Improved Data Construction Methods**: Refining methods for constructing textual aesthetics datasets is crucial for advancing the field.
> 3. **Innovative Training Techniques**: Beyond data improvements, developing new training strategies that prioritize textual aesthetics—such as reinforcement learning with aesthetic feedback or fine-tuning on multimodal datasets—can enhance an LLM's ability to produce aesthetically aligned outputs while maintaining semantic accuracy.
> 4. **Improved Evaluation Frameworks**: Enhancing evaluation frameworks to align more closely with human preferences is critical. Hybrid metrics that integrate semantic clarity, readability, and visual organization can create objective and reliable systems. Robustness checks, such as adversarial testing, will ensure these systems consistently perform across diverse applications.

---

> ### Author Response · Authors · 2024-11-18
> **Rebuttal by Authors (3/4)**
>
> > **Q3: The paper does not clearly explain how to determine if texts with better textual aesthetics are better understood, and how to ensure that human evaluators are making judgments based on content comprehension rather than just neat formatting.**
>
> **A3**: Thank you for your detailed feedback, and we apologize for not providing a more thorough explanation of the cases in the main text due to space constraints. Below, we address your concerns by discussing specific examples from Figure 4 of main text, explaining how to determine if texts with better textual aesthetics are better understood. In the latest version of the paper, we will include the detailed explanation.
>
> Regarding **case 1** at the top of Figure 4 of main text, the example focuses on generating a mnemonic for the Kanji meaning "Wish," using the primitives "clock" and "heart." LLaMA-3.1-8B-TAPO demonstrates significant improvements in textual aesthetics by providing multiple mnemonic options, each clearly separated and thoughtfully worded. This structured presentation enhances the layout and logical organization of the response, making it easier for learners to navigate and choose the mnemonic that resonates most with them. The flexibility offered by multiple options ensures that the text is not only visually appealing but also functionally effective, allowing readers to better understand and retain the information. In contrast, the single mnemonic provided by LLaMA-3.1-8B-Instruct lacks the same level of clarity and variety, making it less engaging and harder to interpret.
>
> Regarding **case 2** in the center of Figure 4, the example examines the text involving the term bug. The response provided by LLaMA-3.1-8B-TAPO enhances readability by emphasizing each occurrence of the term bug with formatting that makes it immediately clear which part of the text each explanation pertains to. This formatting choice allows readers to quickly identify the specific context and meaning of each instance of bug. Additionally, the use of concise phrases aligned with the numbered list structure further improves the layout and clarity of the output. Therefore, **case 2** is easier to understand.
>
> For **case 3** (as previously discussed in **A2**), the folk-style melody output by LLaMA-3.1-8B-Instruct suffers from fragmented line breaks, splitting logical sequences of notes into disjointed segments. This diminishes the readability, visual organization, and coherence of the output, making it challenging for users to interpret and perform the melody accurately. In contrast, LLaMA-3.1-8B-TAPO provides a well-structured output with appropriate line breaks and logical grouping of notes. This improves clarity, layout, and logical structure, making the melody easier to read and more user-friendly for musical performance. Such thoughtful formatting aligns with the principles of textual aesthetics and ensures the usability of the text for its intended purpose.
>
> We hope this explanation helps you better understand how texts with better textual aesthetics can improve readability and comprehension.
>
> > **Q4: How to ensure that human evaluators are making judgments based on content comprehension rather than just neat formatting.**
>
> **A4**: To ensure that human evaluators focus on content comprehension rather than being influenced solely by neat formatting, we implemented the following measures.
>
> **Training and Familiarization:** Before starting the annotation task, evaluators underwent a learning phase to familiarize themselves with the definitions and evaluation criteria for textual aesthetics. They were instructed to comprehensively analyze and compare text samples across the four dimensions (clarity, layout, uniformity, and coherence) to assign scores (win/tie/loss) for textual aesthetics between models. Each annotator was tested on a subset of the dataset to confirm their understanding of these definitions prior to commencing the full annotation task.
>
> **Clear Evaluation Criteria:** we use **well-defined evaluation criteria** aligned with the four dimensions of textual aesthetics: clarity (ease of comprehension), layout (visual organization), uniformity (consistent formatting), and coherence (logical structure). Evaluators are instructed to assess these dimensions holistically while ensuring that content comprehension remains their primary focus.

---

> ### Author Response · Authors · 2024-11-18
> **Rebuttal by Authors (4/4)**
>
> > **Q5: The paper uses GPT-4o for automated assessment, but as a machine, GPT-4o processes only code/symbols/characteristics, and its suitability for evaluating textual aesthetics is questionable.**
>
> **A5**: Thank you for your thoughtful feedback and for raising the concern about the suitability of GPT-4o for evaluating textual aesthetics. Below, we address this question by discussing GPT-4o’s capabilities in understanding textual aesthetics and its consistency with human judgments.
>
> - **GPT-4o's Textual Aesthetic Evaluation Capabilities**
>
>   GPT-4o demonstrates strong text comprehension abilities, allowing it to evaluate textual aesthetics based on the same four dimensions—clarity, layout, uniformity, and coherence—that human evaluators consider. These dimensions are central to assessing textual readability and comprehensibility, as described in our work on Text-Based Text Aesthetic Scoring. By processing text holistically, GPT-4o aligns well with the definitions of textual aesthetics used in our framework.
>   Moreover, **GPT-4o's exceptional multimodal capabilities** enable it to evaluate text aesthetics from a visual perspective as well. This ability allows GPT-4o to bridge both textual and visual aspects of text aesthetics. To further illustrate its multimodal evaluation capabilities, we have included a detailed evaluation case in Appendix F.1.
> - **Consistency with Human Judgments**
>
>   To validate GPT-4o's suitability for evaluating textual aesthetics, we benchmarked its assessments against human. As shown in Figure 2 and Table 5 of main text, the textual aesthetics win rate rankings of our LLaMA-3.1-8B-TAPO and LLaMA-3.1-70B-TAPO models relative to other open-source models are consistent with the win rate rankings obtained using GPT-4o. In our Annotation Consistency analysis, the agreement between GPT-4o and human judges on textual aesthetics preferences, as detailed in Section 6.5, where the TA Text scores demonstrated a 68.67% agreement rate with human annotators and the TA Image scores exhibited a 64.83% agreement rate, indicates that GPT-4o judges have a high concurrence rate with human judges. This agreement rate is comparable to those observed in previous human evaluations, which reported an average of 66% agreement in MT-Bench[1] and 59.7% in UltraFeedback[2]. This alignment demonstrates that GPT-4o reliably captures the defined dimensions of textual aesthetics and provides scores that closely align with human preferences.
>
> ---
>
> **Reference**
>
> [1] Zheng, Lianmin, et al. "Judging llm-as-a-judge with mt-bench and chatbot arena." NeurIPS 2023.
>
> [2] Cui, Ganqu, et al. "Ultrafeedback: Boosting language models with scaled ai feedback" ICML 2024.
>
> ---
> We want to express our sincere gratitude for your review. If you have any further questions or concerns, please feel free to contact us at any time. We are always available and look forward to further discussions with you. :)
>
> Best regards,
>
> All Authors

---

> ### Author Response · Authors · 2024-11-25
> **Official Comment by Authors**
>
> Dear Reviewer NtqG,
>
> We sincerely appreciate your constructive feedback on our manuscript. Guided by your insightful suggestions, we have added the details of human annotation in Section 6.5 and Appendix G, as well as a detailed case explanation in Section 7 to demonstrate how texts with better textual aesthetics are better understood. These updates have been included in the revised manuscript.
>
> Thank you once again for your time and valuable guidance. If you have any further questions or concerns, please feel free to contact us at any time.
>
> Sincerely,
>
> All Authors

---

> > ### Comment · Reviewer_NtqG · 2024-11-25
> >
> > Thank you for the detailed response, which has addressed some of my concerns. However, I have the following two questions regarding your reply:
> >
> > 1. Regarding the clarity (ease of comprehension) of the text, how can we ensure that reviewers genuinely understand the content? I believe a reasonable evaluation approach would involve designing reading comprehension questions based on the content. These could include multiple-choice questions, fill-in-the-blank questions, or short-answer questions to assess the annotators' understanding of the core ideas. This method could help quantify both the time required for comprehension and the accuracy of understanding.
> >
> > 2. As for the example at the bottom of Figure 4 concerning the musical notation, I believe it is inappropriate to explain this issue as a matter of textual aesthetics. The musical notation output by LLaMA-3.1-8B-Instruct constitutes a formatting error. Musical notation without syllable division is akin to sentences without punctuation marks—it is a basic formatting mistake, not a matter of aesthetics.

---

> > > ### Author Response · Authors · 2024-11-25
> > > **Response to Reviewer NtqG**
> > >
> > > Dear Reviewer NtqG,
> > >
> > > Thank you for your response and reviewing our work. We address your feedback point by point below.
> > >
> > > > **Q1**: Regarding the clarity (ease of comprehension) of the text, how can we ensure that reviewers genuinely understand the content? I believe a reasonable evaluation approach would involve designing reading comprehension questions based on the content. These could include multiple-choice questions, fill-in-the-blank questions, or short-answer questions to assess the annotators' understanding of the core ideas. This method could help quantify both the time required for comprehension and the accuracy of understanding.
> > >
> > > **A1:** We sincerely thank the reviewer for your thoughtful feedback and for proposing the use of reading comprehension questions as a method to evaluate text understanding. This approach indeed offers a systematic and quantifiable way to measure comprehension accuracy and processing time, and we acknowledge its value in certain research contexts.
> > >
> > > However, the primary focus of our study is on human preferences regarding textual aesthetics, which are based on an overall perception of the aesthetic quality during the reading process. Human evaluators are capable of identifying which text is easier to understand, and in cases where a distinction cannot be made, they assign a tie, reflecting equivalence in readability. This evaluation method aligns with existing approaches such as MT-Bench[1] and ULtraFeedback[2], which similarly rely on human judgments to assess attributes like instruction following, helpfulness, informativeness, and truthfulness. Notably, evaluating aspects such as instruction following, helpfulness, informativeness, and truthfulness is not easier than assessing readability.
> > >
> > > To ensure the reliability of our evaluations, all annotators participated in standardized training to familiarize themselves with the evaluation criteria and develop a consistent approach to the task. Furthermore, the annotators are proficient in English, with expertise equivalent to that of domain experts, surpassing the capabilities of typical crowd-sourced annotators. This level of proficiency and training ensures that reviewers genuinely understand the content and provide informed and accurate judgments regarding textual aesthetics.
> > >
> > >
> > > ---
> > >
> > > **Reference**
> > >
> > > [1] Zheng, Lianmin, et al. "Judging llm-as-a-judge with mt-bench and chatbot arena." NeurIPS 2023.
> > >
> > > [2] Cui, Ganqu, et al. "Ultrafeedback: Boosting language models with scaled ai feedback" ICML 2024.
> > >
> > >
> > > > **Q2**: As for the example at the bottom of Figure 4 concerning the musical notation, I believe it is inappropriate to explain this issue as a matter of textual aesthetics. The musical notation output by LLaMA-3.1-8B-Instruct constitutes a formatting error. Musical notation without syllable division is akin to sentences without punctuation marks—it is a basic formatting mistake, not a matter of aesthetics.
> > >
> > > **A2:** We appreciate the reviewer’s observation regarding the example of musical notation at the bottom of Figure 4. While it is valid to classify the absence of syllable division in musical notation as a formatting error, we argue that such formatting issues also directly contribute to and overlap with textual aesthetics, particularly in the context of our study.
> > >
> > > Textual aesthetics, as defined in our work, encompass not only the visual organization but also the coherence and clarity of the text. Proper syllable division in musical notation enhances readability and logical flow, enabling users to more intuitively interpret and perform the music. This aligns with our broader framework, where the structural organization of text—including line breaks, spacing, and grouping—serves both functional and aesthetic purposes. The presence or absence of these features influences the user’s experience, blending considerations of formatting and aesthetics.
> > >
> > > In this specific case, the well-structured output by LLaMA-3.1-8B-TAPO, which includes appropriate syllable division and logical grouping of notes, improves the overall usability of the musical notation. This makes it not only functionally accurate but also more visually coherent and pleasing—core aspects of textual aesthetics. Conversely, the fragmented and incoherent output by LLaMA-3.1-8B-Instruct reduces readability and aesthetic appeal, which underscores the importance of aesthetics in functional outputs like musical notation.
> > >
> > > ----
> > > We want to express our sincere gratitude for your review. If you have any further questions or concerns, please feel free to contact us at any time. We are always available and look forward to further discussions with you. :)
> > >
> > > Best regards,
> > >
> > > All Authors

---

> > > > ### Comment · Reviewer_NtqG · 2024-11-27
> > > >
> > > > 1. The authors emphasize that GPT-4 performs comparably to human judgment, aiming to validate its reliability in evaluating textual aesthetics, which I acknowledge and agree with.
> > > >
> > > > 2. However, I am concerned that the assessment of textual aesthetics should not rely solely on subjective judgment. Rigorous objective evaluation is essential. Without such standards, it remains unclear whether textual aesthetics genuinely enhance comprehension or merely improve visual appeal.
> > > >
> > > > 3. If the focus is solely on visual appeal, subjective evaluation might suffice, but this limits the broader significance of textual aesthetics research. For meaningful progress, textual aesthetics should aim to improve human understanding, and demonstrating this requires objective evaluation.
> > > >
> > > > Therefore, I would like to keep the current score.

---

> ### Author Response · Authors · 2024-11-28
> **Response to Reviewer NtqG**
>
> Thank you for your feedback! We sincerely thank the reviewer for your understanding and acknowledgment regarding the reliability of GPT-4o’s performance in evaluating textual aesthetics. Regarding your second and third concerns, We address your feedback point by point below.
>
> > **C1:** However, I am concerned that the assessment of textual aesthetics should not rely solely on subjective judgment. Rigorous objective evaluation is essential. Without such standards, it remains unclear whether textual aesthetics genuinely enhance comprehension or merely improve visual appeal.
>
> During the human evaluation process, annotators are instructed to make comprehensive judgments based on four aspects: readability, layout, uniformity, and coherence. The assessment of whether the content is easier to understand is not merely based on visual appeal but rather on the ease of understanding the corresponding answers to specific questions derived from the text. For example, if Text A contains more headings, line breaks, bold fonts, and other visual elements than Text B, but its content is not easier to understand than that of Text B, then Text A would receive a "loss" label compared to Text B. Conversely, if both texts are equivalent in content but one exhibits better visual organization, making the desired information easier to extract, it would receive a "win" label. This approach ensures that annotators genuinely understand the content and that judgments are not solely influenced by visual appeal.
>
> Our evaluation methodology aligns with those used in prior human preference annotation tasks such as MT-Bench and Arena-Hard, which rely on similar preference-based annotations. The high internal consistency among human annotators and their strong agreement with GPT4 evaluations—comparable to those observed in MT-Bench and Ultrafeedback—validate the reliability of this approach for evaluating textual aesthetics. These results collectively support the adequacy of our method for assessing textual aesthetics and their contribution to comprehension.
>
> > **C2:** If the focus is solely on visual appeal, subjective evaluation might suffice, but this limits the broader significance of textual aesthetics research. For meaningful progress, textual aesthetics should aim to improve human understanding, and demonstrating this requires objective evaluation.
>
> Thank you for raising this important point. As outlined in Rebuttal **A2**, our definition of textual aesthetics explicitly focuses on improving the readability and comprehension of text, rather than solely enhancing its visual appeal. We appreciate your understanding of this core aspect of our research.
>
> As discussed in **C1**, our human evaluation protocol ensures that annotators genuinely assess the content based on its ease of reading and understanding. This is reflected in their scoring, which aligns with the methodology employed in prior studies, demonstrating results comparable to those of established works. Additionally, we have implemented the following measures to ensure the objectivity and fairness of our evaluation process:
>
> - **Standardized Presentation**: All texts were rendered in a uniform visual format, and any information about the originating model was removed to eliminate potential biases.
> - **Independent Judgments**: Annotators performed their evaluations independently, without any communication among them during the process.
>
> These precautions ensure that the evaluations are both rigorous and unbiased. The combination of our well-defined textual aesthetics framework, a robust evaluation protocol, and consistent results substantiates the claim that our methodology effectively assesses textual aesthetics and their contribution to human understanding.
>
> We recognize that other approaches, such as reading comprehension tasks, could further strengthen the evaluation process, and we are actively exploring more objective methods to integrate into future iterations of this research. However, we believe that the current human evaluation setup provides a reliable foundation for assessing textual aesthetics and their broader implications.
>
> ---
> We sincerely hope that the above response addresses your concerns. If you have any further questions or concerns, please feel free to contact us at any time. We are always available and look forward to further discussions with you.
>
> Sincerely,
>
> All Authors

---

### Official Review · Reviewer_sukG · 2024-11-03

**Soundness:** 3
**Presentation:** 2
**Contribution:** 2
**Rating:** 6
**Confidence:** 4

**Summary:**

This paper focuses on textual aesthetics in large language models (LLMs). It first highlights the importance of textual aesthetics, which has been less explored compared to image aesthetics despite the widespread use of LLMs.
Contributions:
1. Dataset Construction:
• Developed an aesthetic data generation pipeline leveraging GPT - 4o for aesthetic polishing.
• Constructed the first aesthetic dataset in the LLM domain, TEXAES, with 50,390 prompts data.
2. Fine - Tuning Method:
• Proposed a textual aesthetics - powered fine - tuning method, TAPO, based on direct preference optimization. It uses the Plackett - Luce model with adjustable optimization weights to better leverage TEXAES and enhance aesthetic fine - tuning performance while preserving general performance.
3. Evaluation Methods:
• Developed two evaluation pipelines for textual aesthetics: one based on text and the other based on images.
• Validated the effectiveness of TEXAES and TAPO through extensive experiments. Fine - tuned LLaMA series models using TEXAES and TAPO and compared their aesthetic scores with state - of - the - art LLMs. Also, employed human experts for professional evaluation. Results showed improvements in aesthetic scores and general capabilities on some benchmarks.

**Strengths:**

Originality
• The paper shows originality in addressing textual aesthetics in LLMs, an area that has received less attention compared to image aesthetics. The construction of the TEXAES dataset and the proposed TAPO fine-tuning method are novel contributions.
Quality
• The research methodology appears to be of good quality. The construction of the dataset through an aesthetic polishing pipeline and the use of appropriate evaluation methods (text-based and image-based) demonstrate a systematic approach.
Clarity
• The paper is generally clear in its presentation. The introduction effectively sets the context and the importance of textual aesthetics. The methods section explains the dataset construction, fine-tuning method, and evaluation pipelines in a understandable manner.
Significance
• The work is significant as it fills a gap in the study of LLMs by focusing on textual aesthetics. The proposed techniques have the potential to improve the aesthetic quality of LLM outputs, which can enhance user experience and the usability of these models in various applications.

**Weaknesses:**

Dataset Limitations
• While the construction of the TEXAES dataset is a significant step, it may have limitations. The dataset is built based on a filtered version of UltraFeedback, and there could be potential biases introduced during this process. For example, the responses in UltraFeedback might already have a certain style or pattern that could limit the diversity of the aesthetic preferences captured in TEXAES.
Evaluation Complexity
• The evaluation methods, although comprehensive with text-based and image-based scoring, could be further refined. The use of GPT - 4o as a judge in both text and image evaluations might introduce some subjectivity and reliance on a single model. There could be a need for more diverse evaluation metrics or a more objective way to combine the text and image evaluations to get a more accurate assessment of textual aesthetics.
Generalizability
• The experiments are mainly focused on the LLaMA series models. It is not clear how well the proposed methods (TEXAES and TAPO) would generalize to other LLMs. There is a need for more extensive experiments across different types of LLMs to demonstrate the broader applicability of the techniques.

**Questions:**

1. Dataset Construction Details:
• Could the authors please provide more details about the filtering process used to create TEXAES from UltraFeedback? How were the responses selected and what criteria were used to ensure a diverse range of aesthetic preferences?
2. Evaluation Metric Objectivity:
• Given that GPT - 4o is used as a judge in both text and image evaluations, how can the authors ensure the objectivity of the evaluation metrics? Are there any plans to explore alternative evaluation methods or to combine multiple evaluation models to reduce subjectivity?
3. Generalizability to Other LLMs:
• The experiments focused on LLaMA series models. What are the authors' expectations or initial findings regarding the generalizability of the proposed methods (TEXAES and TAPO) to other LLMs? Have any preliminary tests been conducted on different models?

---

> ### Author Response · Authors · 2024-11-18
> **Rebuttal by Authors (1/4)**
>
> Thank you for your detailed, helpful feedback. We address your feedback point by point below.
>
> > **Q1:** Dataset Construction Details: Could the authors please provide more details about the filtering process used to create TEXAES from UltraFeedback?
>
> **A1:** Thank you for your thoughtful question regarding the filtering process used to create TEXAES from UltraFeedback. Below, we provide more details about the filtering process used to create TEXAES from UltraFeedback.
>
> **1. Dataset Construction and Initial Filtering**
>
> The foundation of TEXAES is the **UltraFeedback Binarized** dataset, which is a filtered version of UltraFeedback as described in Zephyr[1]. This dataset was constructed using the following process:
>
> - Each prompt in the UltraFeedback dataset includes four model completions from a range of open-source and proprietary models.
> - GPT-4 evaluated each completion based on criteria such as helpfulness and honesty to assign an overall score.
> - The highest-rated completion (based on the overall score) was selected as the **"chosen"** response.
> - One of the remaining three responses was randomly selected as the **"rejected"** response.
>
> **2. Additional Filtering for TEXAES**
>
> Building upon the UltraFeedback Binarized dataset, two additional filtering steps were applied to refine the dataset for TEXAES:
>
> #### **Binary Classification Filtering**
>
> As part of our aesthetic polishing process (described in Sections 3.2 and 5.1), GPT-4o performed a binary classification to determine whether a response required modification to improve readability and comprehensibility.
>
> - Responses identified as already aesthetically satisfactory (a total of 5,858 entries) were filtered out and excluded from further processing.
> - This step ensured that the dataset retained only responses needing enhancement, streamlining subsequent polishing efforts.
>
> #### **Length Control Filtering**
>
> We analyzed the lengths of the filtered responses to identify excessive verbosity or unnatural text characteristics. To address this:
>
> - Outliers in the length distribution (before and after aesthetic polishing) were excluded.
> - Only responses within the 90% confidence interval were retained, as detailed in Appendix A.
>
> To validate the impact of length filtering, we conducted an ablation experiment:
>
> - A model (LLaMA-3.1-8B-Base) was trained on datasets with and without length filtering using DPO($y_t$, $y_l$).
> - Results, detailed in Table 9 of Appendix D, demonstrated that length-filtered data significantly improved model performance across all evaluation tasks. Furthermore, length-filtered responses were shorter, more concise, and easier to read, confirming the effectiveness of this filtering step.
>
> ---
>
> **Reference**
>
> [1]Tunstall, Lewis, et al. "Zephyr: Direct distillation of lm alignment." COLM 2024.

---

> ### Author Response · Authors · 2024-11-18
> **Rebuttal by Authors (2/4)**
>
> > **Q2**  The construction of the TEXAES dataset may have limitations. The dataset is built based on a filtered version of UltraFeedback, and there could be potential biases introduced during this process.  For example, the responses in UltraFeedback might already have a certain style or pattern that could limit the diversity of the aesthetic preferences captured in TEXAES. How were the responses selected and what criteria were used to ensure a diverse range of aesthetic preferences?
>
> **A2:** Thank you for raising this important question and for pointing out the potential limitations in our dataset construction. Below, we address the concerns regarding Response Selection Criteria and Ensuring Diversity of Aesthetic Preferences:
>
> **1. Response Selection Criteria**
>
> Responses in UltraFeedback were selected as follows:
>
> - **Chosen Responses**: For each prompt, the "chosen" responses were selected as the highest-rated completions based on their overall scores, which consider factors like instruction-following, coherence, and helpfulness.
> - **Rejected Responses**: A randomly chosen response from the remaining three completions served as the "rejected" response, ensuring sufficient variability in the dataset.
>
> **2. Ensuring Diversity of Aesthetic Preferences**
>
> - **Diverse Input Prompts in UltraFeedback**: UltraFeedback contains prompts covering a wide variety of domains, topics, and levels of complexity. This inherent diversity provides a strong foundation for constructing the TEXAES dataset, ensuring coverage across different contexts and aesthetic nuances.
> - **Limitations Due to UltraFeedback**: While UltraFeedback offers diverse input prompts, its stylistic tendencies may influence the aesthetic preferences captured in TEXAES. For example, the dataset's original responses may carry specific patterns or stylistic biases, potentially limiting the overall diversity in textual aesthetics. We acknowledge that these constraints stem from our reliance on UltraFeedback as the base dataset.
> - **Generalizability of the Aesthetic Text Polishing Pipeline**: Despite these limitations, the aesthetic text polishing pipeline we developed is highly generalizable and not limited to UltraFeedback. The pipeline can be applied to other instruction datasets, allowing for stylistic and structural variations that expand the aesthetic range of the resulting dataset. This flexibility ensures that TEXAES and similar datasets can adapt to diverse input data sources while maintaining high standards of quality and consistency.
> - **Future Improvements and General Applicability**: To address the limitations and advance TEXAES, our further research includes the following directions:
>
>     - Applying the polishing pipeline to additional datasets to overcome stylistic biases inherent in UltraFeedback.
>     - Refining the pipeline to better adapt to datasets with varying stylistic and content characteristics, further increasing the range of aesthetic preferences captured.
>     - Exploring novel methods to enhance diversity and quality in aesthetic dataset construction, ensuring broader coverage and general applicability for future research.
>
> By combining UltraFeedback's strengths with our generalizable pipeline, TEXAES represents a significant step forward in advancing research on textual aesthetics while maintaining the flexibility to address potential limitations in future iterations.

---

> ### Author Response · Authors · 2024-11-18
> **Rebuttal by Authors (3/4)**
>
> > **Q3:** Evaluation Metric Objectivity: Given that GPT4o is used as a judge in both text and image evaluations, how can the authors ensure the objectivity of the evaluation metrics?
>
> **A3:** Thank you for your thoughtful question regarding the objectivity of the evaluation metrics when using GPT-4o as a judge for both text and image evaluations. Ensuring the fairness and consistency of the evaluation framework is a critical focus of our study. Below, we outline the measures we implemented to ensure objectivity:
>
> 1. **Standardized Baseline for Pairwise Comparisons**
>
>     We employed GPT-4-0314 as a standard baseline for all pairwise comparisons. This consistent reference model eliminated variability in the evaluation process and enabled fair comparisons across models by providing a uniform scoring standard for all win rate and aesthetic evaluations.
>
> 2. **Two-Game Setup to Mitigate Positional Bias**
>
>     To address potential positional bias in pairwise evaluations, we implemented a two-game setup:
>
>     - The positions of the compared models were swapped in separate evaluations.
>     - Results from both comparisons were aggregated using the Bradley-Terry model, a statistically robust ranking method that minimizes positional and systemic biases.
> 3. **Evaluation Across Detailed Aspects**
>
>    GPT-4o’s evaluations for both text-based and image-based methods were conducted using the same detailed aspects of textual aesthetics. Specifically, each text was analyzed based on the four fundamental components: readability、visual organization、consistency、overall structure. This standardized breakdown ensures that all evaluations are conducted uniformly and fairly across all datasets. A detailed example of GPT-4o’s evaluation process for text can be found in Appendix F.1.
> 4. **Consistency with Human Preferences**
>
>     Human evaluation is the gold standard for assessing human’s textual aesthetics preferences. And in our experiment, the textual aesthetics preferences agreement between GPT-4o and human judges, as detailed in Section 6.5, where the TA Text scores demonstrated a 68.67% agreement rate with the human annotators and the TA Image scores exhibited a 64.83% agreement rate, indicates that GPT-4o judges have a high concurrence rate with human judges. This agreement rate is comparable to those observed in previous human evaluations, which reported an average of 66% agreement in MT-Bench[1] and 59.7% in UltraFeedback[2]. These findings suggest that our GPT-4o judges can serve as objective proxies for human preferences in assessing text aesthetics.
>
>
> By employing standardized baselines, mitigating positional bias, evaluating detailed aspects uniformly, and aligning with human preferences, we ensured that GPT-4o serves as a reliable and objective evaluator for textual aesthetics.
>
> ---
>
> **Reference**
>
> [1] Zheng, Lianmin, et al. "Judging llm-as-a-judge with mt-bench and chatbot arena." NeurIPS 2024.
>
> [2] Cui, Ganqu, et al. "Ultrafeedback: Boosting language models with scaled ai feedback" ICML 2024.
>
> > **Q4:** Are there any plans to explore alternative evaluation methods or to combine multiple evaluation models to reduce subjectivity?
>
> **A4:** Thank you for your thoughtful question. We recognize the importance of reducing subjectivity in evaluation metrics and ensuring a comprehensive assessment of textual aesthetics. Below, we outline our ongoing plans to explore alternative methods and combine evaluation mod
>
> - **Development of Hybrid Evaluation Metrics**
>
>   We are working on integrating text-based and image-based textual aesthetic scores into hybrid evaluation metrics. This combined approach will evaluate semantic clarity alongside visual organization, providing a more comprehensive and objective assessment of text aesthetics. By leveraging the complementary strengths of each metric, the hybrid system will offer deeper insights into the overall aesthetic quality of texts.
>
> - **Training a Multimodal Reward Model**
>
>   To align evaluations more closely with human preferences, we plan to develop a multimodal reward model that combines text and image scoring. This reward model will be trained using a diverse set of human annotations, capturing a broader spectrum of aesthetic preferences and reducing reliance on a single evaluation system.
>
> - **Robustness Checks**
>
>   We will introduce robustness checks, such as adversarial testing, to evaluate the consistency and reliability of the metrics. This testing will help identify potential weaknesses and improve the resilience of our evaluation framework.

---

> ### Author Response · Authors · 2024-11-18
> **Rebuttal by Authors (4/4)**
>
> > **Q5:** Generalizability to Other LLMs: The experiments focused on LLaMA series models. What are the authors' expectations or initial findings regarding the generalizability of the proposed methods (TEXAES and TAPO) to other LLMs? Have any preliminary tests been conducted on different models?
>
> **A5:** This is a meaningful suggestion. To investigate the generalizability of our proposed methods (TEXAES and TAPO) to other LLMs, we conducted additional experiments on two different models: Qwen2-7B-Instruct and Mistral-7B-Instruct-v0.3. We used TEXAES as the training dataset and applied TAPO for training under the same settings as LLaMA-3.1-8B-Instruct. The results are summarized in Table A.
>
> Both models demonstrated significant improvements in textual aesthetics and general response capabilities after training with TAPO. These results are consistent with our findings for the LLaMA-3.1 models, providing strong evidence for the generalizability of TEXAES and TAPO across diverse LLM architectures.
>
> In the latest version of the paper, we will include the experimental results and analyses for Qwen2-7B and Mistral-7B, further validating the robustness and applicability of our proposed methods.
>
> **Table A. Performance Evaluation of Qwen2-7B and Mistral-7B Models After Training with TEXAES and TAPO**
>
> | Model | TA Text WR (%) | TA Image WR (%) | AlpacaEval LC WR (%) | Arena-Hard WR (%) | MT-Bench Avg. Score | MMLU 5-shot (%) |
> | --- | --- | --- | --- | --- | --- | --- |
> | **Qwen2-7B-Instruct** | 24.63 | 39.40 | 33.43 | 27.69 | 7.48 | 70.46 |
> | **Qwen2-7B-Instruct + DPO ($y_t$, $y_l$)** | 33.84 | 61.23 | 40.16 | 25.30 | 7.19 | 70.34 |
> | **Qwen2-7B-Instruct + TAPO** | **37.99** | **64.28** | **40.27** | **32.40** | **7.48** | **70.49** |
> | **Mistral-7B-Instruct-v0.3** | 8.26 | 28.90 | 29.87 | 17.13 | 6.59 | 61.52 |
> | **Mistral-7B-Instruct-v0.3 + DPO ($y_t$, $y_l$)** | 25.59 | 54.64 | 36.78 | 20.83 | 6.56 | 61.36 |
> | **Mistral-7B-Instruct-v0.3 + TAPO** | **28.55** | **57.84** | **38.53** | **23.10** | **6.80** | **61.55** |
>
> ----
> We want to express our sincere gratitude for your review. We apologize for the delayed response, as we have been dedicating an extended amount of time to conducting experiments, which has kept you waiting. Please let us know if any of your points were not addressed properly, or if you have any additional questions.
>
> Best regards,
>
> All Authors

---

> ### Author Response · Authors · 2024-11-25
> **Official Comment by Authors**
>
> Dear Reviewer sukG,
>
> We sincerely appreciate your constructive feedback on our manuscript. Guided by your insightful suggestions, we have included experiments in Section 6.1 and Appendix H of the revised manuscript to evaluate the generalizability of TEXAES and TAPO on other LLMs.
>
> Thank you once again for your time and valuable guidance. If you have any further questions or concerns, please feel free to contact us at any time.
>
> Sincerely,
>
> All Authors

---

### Official Review · Reviewer_uWJT · 2024-11-04

**Soundness:** 3
**Presentation:** 2
**Contribution:** 2
**Rating:** 3
**Confidence:** 5

**Summary:**

In this paper, the authors tackle the problem of textual aesthetics in LLMs. To this end, they introduce a new dataset named TEXAES. In addition, they proposed Textual Aesthetics Preference Optimization (TAPO) method. The experiments show that the proposed method performs well on the benchmark datasets.

**Strengths:**

+ The paper is well organized.
+ The problem of textual aesthetics in LLMs is interesting.

**Weaknesses:**

-I have doubts about the textual aesthetics scores. The scores should be decided by human, not by ChatGPT.
-The proposed textual aesthetics-powered training actually aims to predict the scores as close as ChatGPT, not human.
-The authors did mention 3 evaluators, 2 graduate students and one professor. First, the number of evaluators is too small. Second, there is no information about the evaluations, for example, age, background, first language, and expertise.

**Questions:**

Why did the authors use ChatGPT to provide aesthetics scores? Why not human?

Why is there no information about the evaluators? Is it fair for the data collection?

---

> ### Author Response · Authors · 2024-11-18
> **Rebuttal by Authors (1/2)**
>
> We are grateful to the Reviewer for the extensive review. We address your questions point by point below.
>
> > **Q1:** Why did the authors use ChatGPT for aesthetics scores instead of human evaluators?
>
> **A1:** We adopted GPT4o for aesthetics scoring based on both practical and methodological considerations. While human evaluation remains the gold standard for assessing textual aesthetics preferences, it is prohibitively expensive and time-consuming to consistently employ human judges. We chose GPT4o for aesthetics scoring due to its demonstrated consistency and cost-effectiveness in alignment tasks, as seen in related research (e.g., AlpacaEval[1], MT-Bench[2], and Arena-Hard[3]). As shown in Figure 2 of main text, the textual aesthetics win rate rankings of our LLaMA-3.1-8B-TAPO and LLaMA-3.1-70B-TAPO models relative to other open-source models are consistent with the win rate rankings obtained using GPT-4o. In our Annotation Consistency analysis, the agreement between GPT-4o and human judges on textual aesthetics preferences, as detailed in Section 6.5, where the TA Text scores demonstrated a 68.67% agreement rate with human annotators and the TA Image scores exhibited a 64.83% agreement rate, indicates that GPT-4o judges have a high concurrence rate with human judges. This agreement rate is comparable to those observed in previous human evaluations, which reported an average of 66% agreement in MT-Bench[2] and 59.7% in UltraFeedback[4]. These findings suggest that our GPT-4o judges can serve as effective proxies for human preferences in assessing text aesthetics.
>
> ------
>
> **Reference**
>
> [1] Dubois, Yann, et al. "Length-controlled alpacaeval: A simple way to debias automatic evaluators." COLM 2024.
>
> [2] Zheng, Lianmin, et al. "Judging llm-as-a-judge with mt-bench and chatbot arena." NeurIPS 2023.
>
> [3] Li, Tianle, et al. "From Crowdsourced Data to High-Quality Benchmarks: Arena-Hard and BenchBuilder Pipeline." arXiv 2024.
>
> [4] Cui, Ganqu, et al. "Ultrafeedback: Boosting language models with scaled ai feedback" ICML 2024.
>
> > **Q2:** Why is there no information about the evaluators? Is it fair for the data collection?
>
> **A2:** Thank you for your suggestion. We recognize the importance of providing detailed information about the evaluators and their process and we will include human evaluators information in the latest version.
>
> In our study, we employed three annotators: two graduate students in computer science and one professor with a background in applied linguistics. All three evaluators are non-native English speakers but are proficient in English. Their diverse academic and linguistic backgrounds provide a balanced perspective for assessing textual aesthetics across the four key dimensions—clarity, layout, uniformity, and coherence.
>
> Before beginning the annotation process, the evaluators underwent comprehensive training that included case studies and task-specific examples. This training ensured a consistent understanding of the textual aesthetics evaluation criteria and reduced individual interpretation biases. After the training phase, the evaluators were tested on a subset of the dataset to confirm alignment with the evaluation framework and readiness to perform the task.
>
> To ensure fairness and minimize potential biases, the following measures were implemented:
>
> - **Standardized Presentation**: Texts were rendered into a uniform visual format, with all information about the originating model removed. This step ensured that annotations focused purely on textual content and aesthetics without being influenced by the source of the text.
> - **Independent Judgments**: Each annotator performed their evaluations independently, with no communication among annotators during the process. This safeguarded against mutual influence or group bias.
> - **Balanced Sample Distribution**: The dataset provided to annotators included balanced representations of all models under evaluation, preventing skewed exposure to any specific model.
>
> > **Q3**: The number of evaluators is too small
>
> **A3**: In line with previous related works such as ImageReward[1] and UltraFeedback[2] where three evaluators were commonly employed for human studies, we believe that using three annotators provides a sufficient and robust basis to support our conclusions. The use of three evaluators is a standard practice in similar studies, ensuring a balance between reliable annotation quality and practical feasibility.
>
> ---
>
> **Reference**
>
> [1] Xu, Jiazheng, et al. "Imagereward: Learning and evaluating human preferences for text-to-image generation." NeurIPS 2024.
>
> [2] Cui, Ganqu, et al. "Ultrafeedback: Boosting language models with scaled ai feedback"  ICML 2024.

---

> ### Author Response · Authors · 2024-11-18
> **Rebuttal by Authors (2/2)**
>
> > **Q4**: The proposed textual aesthetics-powered training actually aims to predict the scores as close as ChatGPT, not human.
>
> **A4:** Our proposed textual aesthetics-powered training is designed to improve the responses of large language models by enhancing their readability and comprehensibility, ensuring the outputs are easier to read, understand, and interact with. The ultimate goal is to align the models’ responses with human textual aesthetics preferences.
>
> To achieve this, we employ GPT-4o as a cost-effective and scalable evaluator for text aesthetics. As shown in Figure 2 and Table 5 of the main text, GPT-4o’s Text-Based and Image-Based Text Aesthetic Scoring methods demonstrate considerable consistency with human textual aesthetics preferences. This consistency indicates that GPT-4o can effectively reflect human-like scoring tendencies in textual aesthetics evaluations.
>
> By leveraging GPT-4o, we are able to process large datasets efficiently, ensuring that the training aligns with human preferences while maintaining scalability. While GPT-4o is used as an evaluator in our framework, the ultimate aim remains rooted in improving the aesthetics of LLM outputs to meet human textual preferences.
>
> ---
> We want to express our sincere gratitude for your review. If you have any further questions or concerns, please feel free to contact us at any time. We are always available and look forward to further discussions with you. :)
>
> Best regards,
>
> All Authors

---

> > ### Comment · Reviewer_uWJT · 2024-11-25
> >
> > The authors have attempted to address the raised questions. However, I believe the paper still has significant issues regarding the use of GPT-4o for aesthetic scoring, as well as concerns about the consistency of the human evaluators. These issues were also highlighted by another reviewer. Therefore, I would like to keep my original score.

---

> > > ### Author Response · Authors · 2024-11-28
> > > **Response to Reviewer uWJT**
> > >
> > > Thank you for your feedback! We appreciate your suggestion, and we have added the details of human annotation in Section 6.5 and Appendix G. Regarding your concerns, we address your feedback point by point below.
> > > >**C1:**  Use of GPT-4 for Aesthetic Scoring
> > >
> > >  As outlined in our previous response (**A1**), we selected GPT-4 for its demonstrated consistency, scalability, and cost-effectiveness, making it a practical choice for large-scale aesthetic evaluations. The consistency between GPT-4o and human evaluators, as detailed in the Annotation Consistency analysis (68.67% agreement on textual aesthetics preferences), supports GPT-4's ability to effectively mirror human aesthetic preferences. This level of agreement is consistent with similar studies, such as MT-Bench[1] (66%) and UltraFeedback[2] (59.7%). These results validate the reliability of using GPT-4o for evaluating textual aesthetics.
> > >
> > > >**C2:** Consistency of Human Evaluators
> > >
> > > As described in Section 3.3, textual aesthetics are evaluated across four fundamental dimensions: clarity (ease of comprehension), layout (visual organization), uniformity (consistent formatting), and coherence (logical structure). These dimensions are assessed together, resulting in a comprehensive Likert-scale evaluation (1–5) during pairwise comparisons against a baseline model (GPT-4-0314). For robustness, the Bradley-Terry model is utilized to aggregate these pairwise comparisons into a unified ranking, ensuring consistency and quantitative evaluation of textual aesthetics.
> > >
> > > To ensure consistency among human evaluators, we implemented a rigorous preparation process. Before starting the annotation task, evaluators underwent a learning phase to familiarize themselves with the definitions and evaluation criteria for textual aesthetics. They were instructed to comprehensively analyze and compare text samples across the four dimensions (clarity, layout, uniformity, and coherence) to assign scores (win/tie/loss) for textual aesthetics between models. Each annotator was tested on a subset of the dataset to confirm their understanding of these definitions prior to commencing the full annotation task.
> > >
> > > Additionally, consistency among evaluators was verified through an inter-annotator agreement study (reported in Section 6.5, Table 5), which showed an average agreement rate of 78.84% among annotators which is comparable to those observed in previous human evaluations consistency(MT-Bench[1], Ultrafeedback[2]), confirming that evaluators consistently applied the evaluation criteria. This process ensures reliable and consistent textual aesthetics assessment.
> > >
> > > ---
> > >
> > > **Reference**
> > >
> > > [1] Zheng, Lianmin, et al. "Judging llm-as-a-judge with mt-bench and chatbot arena." NeurIPS 2023.
> > >
> > > [2] Cui, Ganqu, et al. "Ultrafeedback: Boosting language models with scaled ai feedback" ICML 2024.
> > >
> > > ---
> > > We sincerely hope that the above response addresses your concerns. If you have any further questions or concerns, please feel free to contact us at any time. We are always available and look forward to further discussions with you.
> > >
> > > Sincerely,
> > >
> > > All Authors

---

### Author Response · Authors · 2024-11-27
**General Response by Authors**

We sincerely thank the reviewers for their thorough feedback and insightful comments.

We have carefully addressed each point raised and incorporated all constructive suggestions into the revised manuscript, with modifications highlighted in blue text.

Should any aspects require further clarification, we welcome additional questions.

---

### Note · Authors · 2024-12-13

I have read and agree with the venue's withdrawal policy on behalf of myself and my co-authors.